# Neutrophil azurophilic granule glycoproteins are distinctively decorated by atypical pauci- and phosphomannose glycans

Karli R. Reiding [1,2,4✉], Yu-Hsien Lin[1,2,4], Floris P. J. van Alphen[3], Alexander B. Meijer[1,3] & Albert J. R. Heck [1,2✉]

While neutrophils are critical first-responders of the immune system, they also cause tissue damage and act in a variety of autoimmune diseases. Many neutrophil proteins are *N*-glycosylated, a post-translational modification that may affect, among others, enzymatic activity, receptor interaction, and protein backbone accessibility. So far, a handful neutrophil proteins were reported to be decorated with atypical small glycans (paucimannose and smaller) and phosphomannosylated glycans. To elucidate the occurrence of these atypical glycoforms across the neutrophil proteome, we performed LC-MS/MS-based (glyco)proteomics of pooled neutrophils from healthy donors, obtaining site-specific *N*-glycan characterisation of >200 glycoproteins. We found that glycoproteins that are typically membrane-bound to be mostly decorated with high-mannose/complex *N*-glycans, while secreted proteins mainly harboured complex *N*-glycans. In contrast, proteins inferred to originate from azurophilic granules carried distinct and abundant paucimannosylation, asymmetric/hybrid glycans, and glycan phosphomannosylation. As these same proteins are often autoantigenic, uncovering their atypical glycosylation characteristics is an important step towards understanding autoimmune disease and improving treatment.

[1] Biomolecular Mass Spectrometry and Proteomics, Bijvoet Center for Biomolecular Research and Utrecht Institute for Pharmaceutical Sciences, University of Utrecht, Utrecht, The Netherlands. [2] Netherlands Proteomics Center, Utrecht, The Netherlands. [3] Department of Molecular and Cellular Hemostasis, Sanquin Research, Amsterdam, The Netherlands. [4] These authors contributed equally: Karli R. Reiding, Yu-Hsien Lin. ✉email: k.r.reiding@uu.nl; a.j.r.heck@uu.nl

Neutrophils are the most abundant white blood cells and play a critical role in the immune system, being some of the first responders to migrate towards sites of inflammation and perform local antimicrobial functions[1,2]. Underlining their importance, neutrophils are produced in massive quantities and end up filling more than half the population of leucocytes in human blood—even though the individual cells are relatively short-lived, with lifetimes in the order of days[3]. Neutrophils perform their action via a number of mechanisms, including the phagocytosis of pathogens and subsequent digestion in intracellular phagosomes, the ensnaring of pathogens by releasing DNA in the form of neutrophil extracellular traps (NETs), or by the release of toxic proteins mixtures into the extracellular space in a process called degranulation[4]. This degranulation can occur from several distinct granules, including the azurophilic (or primary) granules, the specific (or secondary) granules and the gelatinase (or tertiary) granules, each loaded with a particular set of proteins[5,6]. Azurophilic granules, for instance, are known to contain the abundant peroxidase myeloperoxidase as well as the proteases neutrophil elastase and myeloblastin (proteinase 3), while the iron-scavenger lactotransferrin and glycosidase lysozyme C are primarily contained within the specific (or secondary) granules[5,6].

While neutrophils are essential to protect against invading pathogens, the cells are unfortunately not very discriminatory once their defences have been activated. Next to handling pathogens, the degranulate inadvertently destroy otherwise healthy proteins and carbohydrates[7], leading to lasting damage to the extracellular matrix and tissue and contributing to disease phenotypes such as asthma and chronic obstructive pulmonary disease (COPD)[8]. Notably, a subset of neutrophil proteins also tends to develop into the target of an autoimmune response, initiated by the collectively named anti-neutrophil cytoplasmic autoantibodies (ANCAs)[9–11]. These ANCAs are found in multiple autoimmune diseases, including rheumatoid arthritis and inflammatory bowel disease, but they are most notable for their diagnostic value in distinguishing subtypes of ANCA-associated vasculitides (AAV)[9–11]. While neutrophils are of obvious clinical interest in terms treating disease complications and uncovering epitopes of autoantigenicity, very little is actually known about the post-translational modifications (PTMs) that naturally occur on the various proteins in neutrophils and/or secreted by them.

One frequent PTM, glycosylation, represents the modification of proteins with one or more complex sugars, which typically exhibit a large degree of functionally relevant variation[12]. Glycosylation has a major effect on a protein as the glycans affect, among other things, the folding and receptor interaction of the protein, as well as the availability of the peptide backbone to proteases and the recognition by autoantibodies[12–15]. On a released glycan level neutrophil glycosylation has been previously characterized[16], as well as after subcellular fractionation thereof[17]. However, the glycosylation characteristics of individual neutrophil glycoproteins are not well-known, aside from detailed specific reports on neutrophil elastase, cathepsin G, and myeloblastin[18–20], and from an early N-glycoproteomics experiment on (neutrophil-rich) sputum covering 115 N-glycopeptides[21,22]. Interestingly, the overall indication from glycoproteomics efforts is that a subset of neutrophil proteins likely contains an in vertebrates infrequently observed a class of small carbohydrates—the relatively short paucimannose glycans[21,23]. Adding to this, our group recently performed a detailed mass spectrometric characterization of the abundant neutrophil protein myeloperoxidase[24]. By this, we uncovered that this glycoprotein indeed harboured, next to the aforementioned paucimannose glycans, a significant amount of phosphomannosylation (Asn323) and highly asymmetric glycan species (Asn483), albeit only specifically on a few of the five N-glycosylation sites[24]. These data were

subsequently confirmed by others[25]. The question remains if these atypical glycosylation traits are common for all neutrophil proteins, or of just a subset thereof. Recent glycomics (released glycan) evidence points towards the azurophilic granules to contain at least paucimannose species, but phosphomannose glycans remain challenging to localize and the involved protein carriers remain to be elucidated[17]. Therefore, next to exploring the overall glycosylation of neutrophil proteins in a site-specific manner, we aimed to confirm their atypical glycan characteristics and to elucidate whether this could be attributed to certain protein subsets.

As such, by means of the latest advancements in sample preparation, hybrid mass spectrometry (MS), and data analysis[26–28], we present an in-depth site-specific characterization of the neutrophil N-glycoproteome. Starting from a pool of neutrophils mixed from 10 independent donors, we used MS-based detection and label-free quantification to obtain data on 2726 intact N-glycopeptides mapping to ~240 neutrophil glycoproteins. By this, we found that neutrophil membrane proteins were mostly decorated by either high-mannose glycans or complex glycans with high levels of fucosylation, likely Lewis-X (CD15) and sialyl-Lewis X (CD15s), while the neutrophil secretory proteins mostly displayed complex glycans. Notably, we found several neutrophil glycoproteins abundantly decorated with paucimannose, phosphomannose and asymmetrical glycans, with a common denominator that all these proteins typically reside in the azurophilic granules. From this, we speculate that the glycosylation machineries of neutrophilic white blood cells may vary through compartmentalization. We believe our data will aid in understanding neutrophil biology and the pathogenicity that may arise from it, as well as informing on leads for targeted intervention within neutrophil-based malignancies.

## Results

To uncover the site-specific N-glycosylation characteristics of neutrophil proteins (Fig. 1a), we pooled the neutrophils from 10 presumably healthy donors to perform in-depth proteomics and glycoproteomics analyses. Cells were lysed and the proteins thereof reduced, alkylated and proteolytically digested by GluC and trypsin. We targeted the glycopeptide by using two complementary approaches: one part of the resulting peptides was desalted by reversed-phase (RP)-solid-phase enrichment (SPE) only, whereas another part was, next to desalting, also enriched for glycopeptides by hydrophilic-interaction liquid chromatography (HILIC)-SPE (Fig. 1b).

**Neutrophil proteomics**. To get a glance at the neutrophil proteome we initially performed shotgun proteomics on the desalted neutrophil (glyco)peptides and analysed them by RP–LC–MS/MS. The resulting MS data were searched against the reviewed human proteome (Swiss-Prot database, release date September 2019) with two different strategies: either searching only for (1) protease-specific peptides allowing up to three miscleavages, or (2) allowing complete non-specificity in proteolytic cleavage.

The more specific search yielded the label-free quantification (LFQ) of 505 proteins detected in at least two out of three replicates and identified by at least three unique peptides, the false-discovery rate being controlled at 1%. Abundant examples hereof included the common neutrophil proteins lactotransferrin and myeloperoxidase, as well as protein S100-A8 and A9 which typically heterooligomerise into calprotectin (Supplementary Data 1)[29]. The identified proteome was in congruence with those from other neutrophil proteomics studies and indicated a broad detection of proteins from different cellular compartments[5,29,30]. From the literature, we could infer that our detections included proteins from the membrane, cytosol,

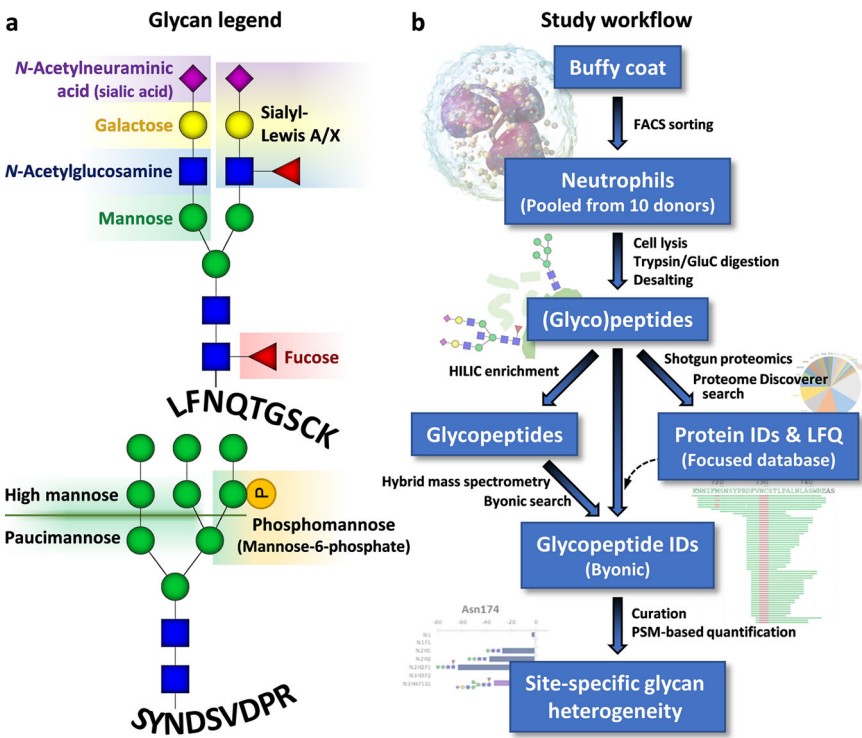

**Fig. 1 Background information for glycan nomenclature and study design. a** Annotations is used to describe glycan compositions. Monosaccharides are represented as: fucose (Fuc; red triangle), mannose (Man; green circle), galactose (Gal; yellow circle), *N*-acetylglucosamine (GlcNAc; blue square), *N*-acetylneuraminic acid (NeuAc; dark magenta diamond), and phosphomannose (phospho; orange circle). **b** Scheme of the experimental workflow. Neutrophils acquired by flow cytometry of buffy coats were pooled from 10 independent, presumably healthy, donors. The cells were disrupted, the released proteins digested, and the peptides were desalted by reversed-phase (RP) solid-phase enrichment (SPE). The resulting (glyco)peptides were then either further enriched by hydrophilic-interaction liquid chromatography (HILIC) SPE before mass spectrometry (MS) analysis, or directly analysed by MS without enrichment. Results from shotgun proteomics (desalting-only) were used for the identification and quantification (LFQ) of the proteins, whereas the glycoproteomics data resulting from both desalting-only and desalting + HILIC were used for identification and quantification of the glycopeptides.

azurophilic (primary) granules, specific (secondary) granules, gelatinase (tertiary) granules, and secretory vesicles[5].

Neutrophil granules are known to harbour a wide selection of proteases to facilitate antimicrobial activity, examples being neutrophil elastase, myeloblastin and cathepsin G. This means that by processing in the cells and during sample preparation a wider variety of peptide cleavages can occur in neutrophil samples beyond those generated with just GluC and trypsin[30]. To account for this, we additionally searched the proteome with no constraint on the peptide sequence, which substantially enlarged the covered proteome and allowed the detection of 4453 proteins (≥2 out of three replicates, ≥3 unique peptides, 1% false-discovery rate) (Fig. 2, Supplementary Data 2). While we did not benchmark label-free quantification for non-tryptic peptides—differences in charge affinity and distribution, fragmentation propensity and interpretability by search engines each demand further investigation—this substantial enlargement of the detectable proteome by using a non-specific search strategy indicates that endogenous processing should be accounted for when analysing neutrophils. The observation of the high frequency and abundance of non-tryptic peptides are in line with previous proteomics studies on primary leucocytes and tumour cells infiltrated by neutrophils[31,32].

**Neutrophil glycoproteomics.** We next subjected the neutrophil (glyco)peptides resulting from both desalting-only and desalting followed by HILIC enrichment to triplicate RP–LC–MS/MS analysis with parameters optimized for glycoproteomics. In particular, this meant that precursor signals were either fragmented by higher-energy collisional dissociation (HCD), or by electron-transfer higher-energy collisional dissociation (EThcD) triggered on glycan fragments from preceding HCD fragmentation. In doing so, we obtained clear MS/MS evidence for the presence of high-mannose glycans, sialylated glycans, as well as high degrees of antennary fucosylation across the glycoproteome (Fig. 3, Supplementary Fig. 1). Importantly, we could also establish the presence of paucimannose and phosphomannose-containing glycan species (Fig. 3a, b).

The acquired glycoproteomics data were searched for the presence of glycopeptides using Byonic[33]. We opted to perform the search without constraints on the proteolytic cleavage, considering our observations from the shotgun proteomics experiments and the evidence that the structure of a glycan may be of large influence to proteolytic efficiency[14]. To retain a manageable search space we filtered our protein databases for those protein sequences that contained at least one *N*-glycosylation sequon (Asn-Xxx-Ser/Thr; Xxx ≠ Pro). Doing so, we searched one instance of each sample type (desalting-only HCD, desalting-only EThcD, desalting + HILIC EThcD) against the full human glycoproteome (Swiss-Prot), and performed searches against the triplicates with a shortlist, considering only the top 500 most abundant glycoproteins as established from the shotgun proteomics results. The majority of peptides proved to be semi-specific for GluC/trypsin digestion, with common alternative cleavage sites occurring at Val, Phe, Leu/Ile, and Tyr (Supplementary Fig. 2).

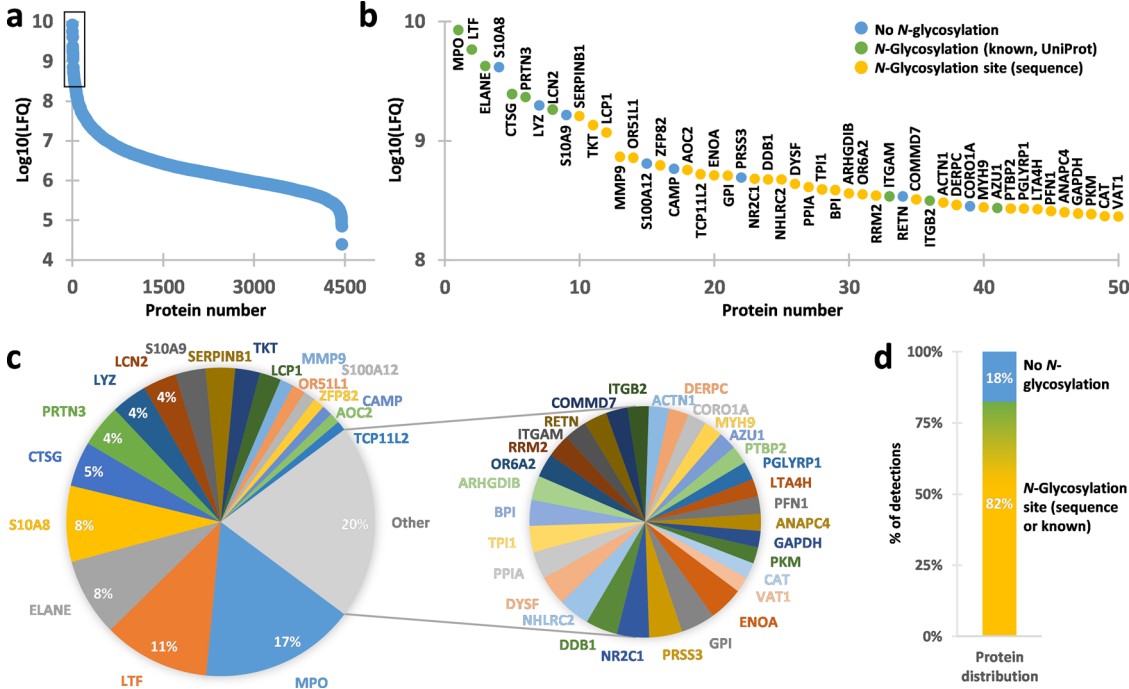

**Fig. 2 Quantitative human neutrophil proteome. a** Quantification of 4453 neutrophil proteins detected by LC-MS/MS after lysis of whole neutrophils and subsequent digestion by trypsin and GluC. Proteins were only included into this quantification when detected in at least two out of three replicates with a minimum of three unique peptides. **b** Close-up of the top 50 most abundant proteins (encapsulating approximately 65% of the total proteome intensity). The colours indicate whether the known protein sequence contained either no N-glycosylation site (blue; sites being defined as Asn-Xxx-Ser/Thr where Xxx ≠ Pro), at least one site with evidence for N-glycosylation as annotated in UniProt (green), or at least one site predicted by sequence analysis (yellow). **c** Relative abundance of the top 50 most abundant proteins. **d** N-Glycosylation site distribution across all detected proteins (nonspecific).

In terms of glycosylation, we allowed for 279 possible N-glycan compositions in the search (Supplementary Data 3), which did encompass also paucimannose species and smaller (e.g., $HexNAc_1dHex_1$ and $HexNAc_2Hex_2$), monoantennary and hybrid species (e.g., $HexNAc_3Hex_3NeuAc_1$ and $HexNAc_3Hex_6$), complex multiantennary species with various degrees of fucosylation (e.g., $HexNAc_4Hex_5NeuAc_2$ and $HexNAc_6Hex_7dHex_5NeuAc_1$), and phosphomannosylated glycans typically originating from the lysosomal pathway of protein degradation (e.g., $HexNAc_2Hex_5Phospho_1$ and $HexNAc_4Hex_8Phospho_2$)[27,34]. Following previously reported data curation parameters[35], we retained a total of 2726 unique N-glycopeptide identifications, informing on the glycosylation of 241 neutrophil proteins with at least one N-glycosylation sequon (Supplementary Data 4). We then assigned global glycosylation characteristics to the glycopeptides on the basis of monosaccharide compositions. We defined the following categories: (1) unoccupied (no monosaccharides on a given site), (2) paucimannose ($HexNAc < 3$ and $Hex < 4$), (3) phosphomannose ($Phospho > 0$), (4) high-mannose ($HexNAc = 2$ and $Hex > 3$), (5) hybrid/asymmetric ($HexNAc = 3$), (6) diantennary ($HexNAc = 4$) and (7) extended ($HexNAc > 4$).

Importantly, the desalting-only and desalting + HILIC enrichment methods yielded complementary information being both somewhat biased to different parts of the neutrophil glycoproteome. As derived from the distribution of peptide spectrum matches (PSMs; the measure of fragmentation spectra that have likely been correctly assigned), the desalting-only method facilitated the detection of paucimannose species across the proteome, whereas such glycopeptides were almost fully lost with subsequent HILIC enrichment (Fig. 4). The additional HILIC step, on the other hand, improved the detection of glycopeptides so that the more analytically challenging higher antennarity compositions became detectable and quantifiable. Due to the

substantial bias observed in desalting + HILIC, we considered the desalting-only method to be leading in the detection of peptide glycoforms. However, as desalting + HILIC provided more information on the (individually) lower abundant high-complexity glycoforms, we kept both methods for analysis and visualization.

In general, neutrophil glycoproteins were mostly decorated with high-mannose, diantennary or extended-type glycosylation (Fig. 5). By inferring the likely cellular locations for these proteins from the literature, we could predict their origins to be either from the cell surface, from within the cell (including the endoplasmic reticulum and Golgi apparatus), from specific and gelatinase granules, from secretory vesicles, and to a lesser degree from azurophilic granules[5]. Neutrophil glycoproteins extensively decorated with a wide variety of glycans included lactotransferrin, neutrophil gelatinase-associated lipocalin, integrin alpha-M (CD11b), integrin beta-2 (CD18), carcinoembryonic antigen-related cell adhesion molecule 8 (CD66b), olfactomedin-4, and many others. Interestingly, for proteins predicted to originate from the azurophilic granules we detected the abundant presence of a quite distinct glycosylation characteristic, namely of partial site-unoccupancy, paucimannose species, hybrid-/asymmetric-type glycans, as well as extensive phosphomannosylation. Proteins carrying these atypical glycosylation features included neutrophil elastase, cathepsin G, myeloblastin (proteinase 3), azurocidin, and myeloperoxidase.

**Site-specific glycan heterogeneity.** We assessed the glycan distribution of each N-glycosylation site on each of the major neutrophil glycoproteins (Fig. 6) by comparing relative PSM numbers, which notably earlier yielded highly similar quantification as manual $MS^1$ area integration (Supplementary Fig. 3). Interestingly, even within glycoproteins distinct glycosylation sites still displayed

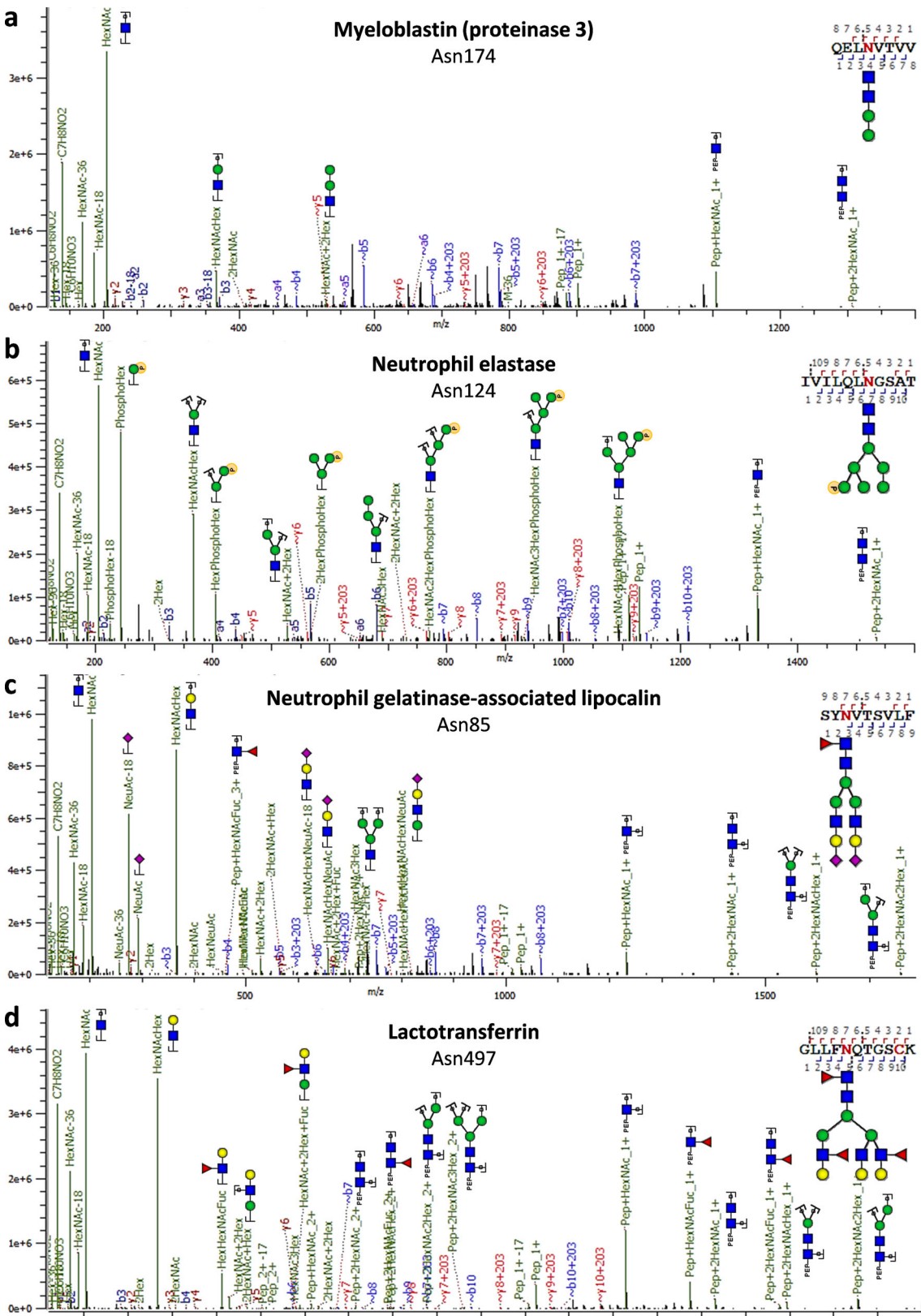

**Fig. 3 Prototypical MS/MS spectra exemplifying the different categories of glycosylation observed in neutrophils. a** Paucimannosylation, **b** phosphomannosylation, **c** sialylation, and **d** high degrees of (antennary) fucosylation. In **a** the fragments b4 + 203 and y5 + 203 position the paucimannosylated glycan unambiguously at the Asn residue, excluding that this peptide carries an O-glycan instead. Similarly, in **b** the phosphorylation clearly is associated with the hexose residues, excluding that is rather on the Ser or Thr residue. The glycoprotein and specific site harbouring the modification is shown on the top of the spectrum. Fragmentation was achieved with higher-energy collisional dissociation (HCD).

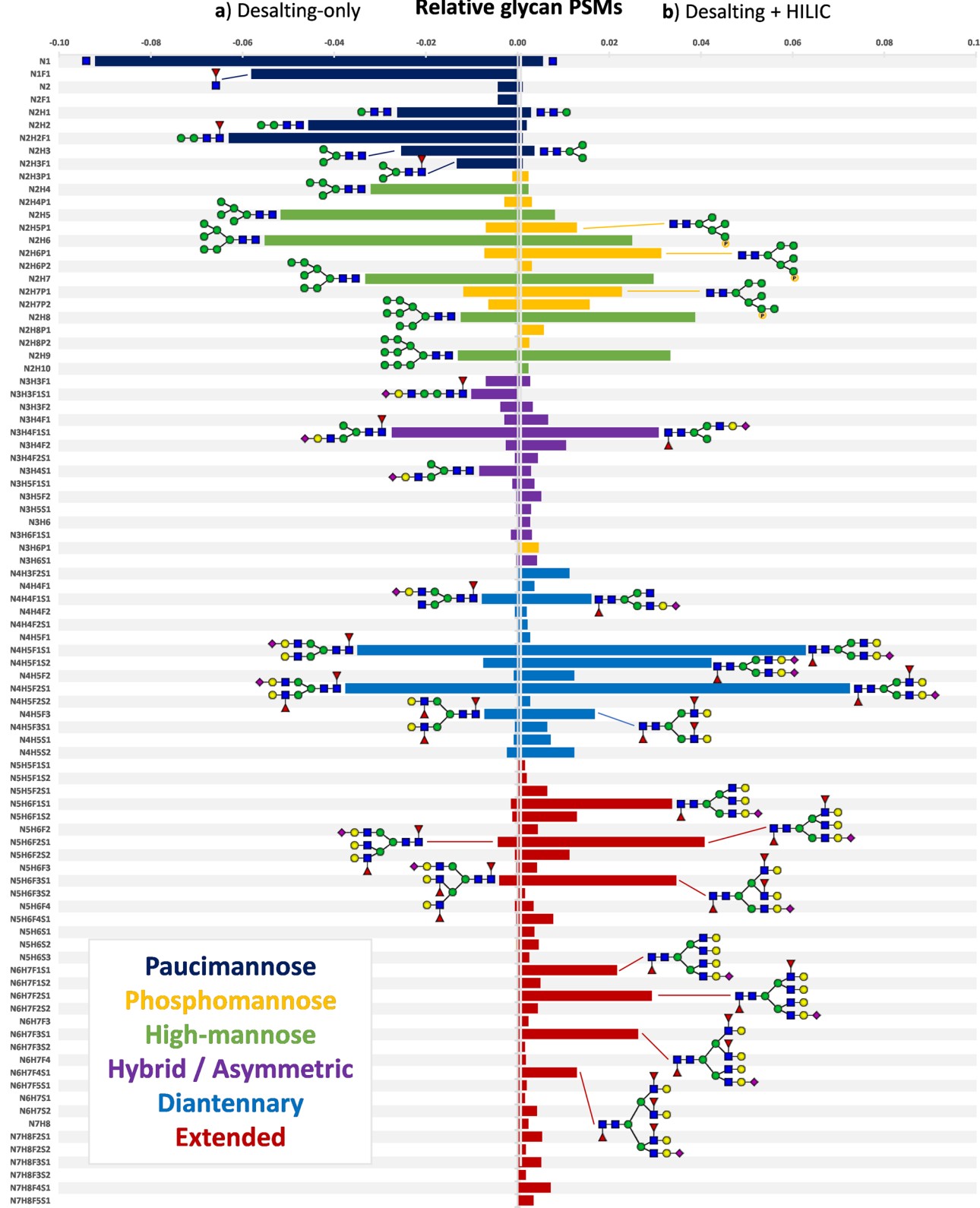

a) Desalting-only **Relative glycan PSMs** b) Desalting + HILIC

Legend: Paucimannose, Phosphomannose, High-mannose, Hybrid / Asymmetric, Diantennary, Extended

unique glycosylation patterns, a concept recently termed meta-heterogeneity[36]. For instance, lactotransferrin carried on its Asn156 exclusively extended glycans (tri- and tetraantennary species with a high degree of antennary fucosylation), while Asn497 expressed diantennary and smaller glycans. Lactotransferrin Asn642 appeared mostly unoccupied, matching previous observations that did not rely on MS for detection[37]. In general, high-mannose and

diantennary/extended species were typically detected on different sites. Integrin alpha-M (CD11b) provides another example, harbouring high-mannose glycan species at sites Asn391, Asn734, Asn801, Asn880, Asn900 (also hybrid), Asn940, Asn978, Asn1021, Asn1044 and Asn1050, but mainly diantennary/extended species at sites Asn240, Asn692, Asn993, and Asn1075. Another example of this meta-heterogeneity was observed for integrin beta-2 (CD18),

**Fig. 4 Quantitative overview of the peptide glycoforms detected across the 500 most abundant neutrophil glycoproteins. a** The relative number of glycopeptides detected by MS (PSMs) directly after desalting (left). **b** The relative number of glycan PSMs detected after sequential desalting and HILIC enrichment (right). As seen from the distributions, HILIC enrichment after desalting leads to a bias against the smallest glycan species but allows for better detection of the lower-abundant higher-complexity glycans. As such, we considered RP to be leading in reporting the glycan distribution of a given glycosylation site but utilized the HILIC data when little or no detections were made with desalting-only. Colours indicate the classes of the glycans, thereby distinguishing paucimannose (dark blue; three mannoses or less), phosphomannose (yellow; high-mannose but at least one phosphorylation), high-mannose (green; four mannoses or more), asymmetric/hybrid (purple; one antennary GlcNAc), diantennary (light blue; two antennary GlcNAcs) and extended species (red; three antennary GlcNAcs or more). Shorthand compositional annotation on the y-axis follows N = N-acetylhexosamine (e.g., N-acetylglucosamine), H = hexose (e.g., mannose or galactose), F = deoxyhexose (fucose), S = sialic acid (N-acetylneuraminic acid), and P = phosphorylation (e.g., mannose-6-phosphate).

on which mainly high-mannose glycans occupied sites Asn501 and Asn636, while diantennary/extended species occupied Asn50, Asn116, and Asn212.

Of special note, for the azurophilic granule proteins we detected altogether different glycosylation characteristics than for proteins originating from other neutrophil locations, but we could still appreciate the large meta-heterogeneity between individual sites. For example, on myeloperoxidase, we detected phospho- and high-mannose glycosylation at Asn323, pauci- to high-mannose species at both Asn355 and Asn391, hybrid/asymmetric/diantennary species at Asn483, and paucimannose glycans together with partial occupancy at Asn729. Another example, neutrophil elastase, exhibited pauci- and phosphomannosylation at Asn88 and diantennary species at Asn124, while Asn173 remained mostly unoccupied. Even for proteins that generally displayed the same glycosylation characteristics across sites, e.g., myeloblastin (proteinase 3) which was primarily paucimannosylated, we still could detect clear differences in which glycan composition was dominant, e.g., $HexNAc_1dHex_1$ at Asn129 and $HexNAc_2Hex_2dHex_1$ at Asn174.

## Discussion

While neutrophils are some of the most critical first responders of our immune system, these cells can also be responsible for much tissue damage partly through the production/secretion of proteins that end up being targets of autoantibodies[9–11]. Given this, it is important to know in which proteoforms the involved proteins actually exist, not only in terms of amino acid sequence but also in terms of their modifications. Neutrophil protein PTMs remains an understudied subject, even while these may have a significant influence on enzymatic activity, interaction with other proteins, inhibitors and scavengers, and the overall accessibility of immunogenic epitopes. Although only a handful reports are available, as reviewed recently[22], it has been hypothesized that released neutrophil proteins may carry N-glycosylation variants that are rarely observed on extracellular proteins within humans, namely paucimannosylation and phosphomannosylation[18–21,24]. As such, the goals in the current study were to (1) verify whether such atypical glycosylation is observed on more, or even all, neutrophil proteins, (2) to establish on which proteins and protein groups these can be observed, and (3) to explore the general glycosylation characteristics of neutrophil proteins in a site-specific manner. By performing mass spectrometric (glyco)proteomics on the neutrophils pooled from 10 apparently healthy donors, we here provide an in-depth investigation into the site-specific glycosylation characteristics of the neutrophil glycoproteome.

Several methodological aspects of our study need to be kept in mind before interpreting the results. Given that neutrophils abundantly produce proteases like neutrophil elastase, cathepsin G and myeloblastin, it came as no surprise that our neutrophil (glyco)proteomics study revealed the formation and identification of many semi-specific and non-specific peptides. Such a plethora of different (glyco)peptides somewhat hamper identification and also

quantification. For the relative quantification of glycan distribution per glycosylation site others and we would typically use manual integration of $MS^1$ areas for individual glycopeptides (e.g., by Skyline)[24,38], but the observed extreme number of glycan and peptide combinations prohibited this approach from a practical perspective. One alternative form of quantification can come from the counting of PSMs (annotated $MS^2$)[27,39]. It can be reasoned that, compared to low-abundant analytes, high-abundant analytes will have longer elution times, more detectable charge states, more detectable structural isomers, more detectable peptide miscleavages, a higher chance of $MS^2$-triggering, as well as a higher chance of identification when triggered, each of these contributing to a higher number of PSMs. Validating this approach on the data we previously generated for the in-depth profiling of isolated neutrophil myeloperoxidase[24], we were pleased to see an excellent approximation of relative glycan distribution per site (Supplementary Fig. 3). As such, while not a full substitute for manual integration, PSM counting may provide a reasonable alternative for complex glycoproteomics samples of this kind.

Sample preparation with either desalting-only or desalting followed by HILIC-SPE presented notably different views of the neutrophil glycoproteome. HILIC is commonly used for the enrichment of glycans and glycopeptides and its retention mechanisms are primarily based on capturing the glycan portion of a glycoconjugate while reversed-phase SPE is typically dependent on the peptide portion[40]. As such, our number of glycopeptide PSMs was far greater in the HILIC method than in RP-only. This also led to the clear detection of analytically complex glycans which might otherwise be missed, e.g., due to splitting of signal intensity across related glycans and glycan isomers. On the other hand, HILIC enrichment also led to the major loss of glycopeptides with paucimannose glycans and smaller, which is understandable because the carbohydrate portion required for HILIC is very small on these peptides. While we here describe paucimannose glycosylation to be "atypical" and "uncommonly observed", this might also be due to a general underestimation across the literature due to the ubiquity of HILIC and lectin-based enrichment for glycan analysis[40–42]. In the work presented here, we interpreted the desalting-only method as dominant for informing on the glycan distribution for a given glycosylation site but relied on the HILIC data when little or no detections were otherwise achieved. Although likely we still miss some glycopeptides, these two methods appeared to be quite complementary in our hands. Other enrichment methods could still prove valuable for exploring different parts of the neutrophil glycoproteome. For example, analysis of phosphomannosylated glycoproteins and glycopeptides could be improved by making use of $Fe^{3+}$-IMAC enrichment, although this would be at the expense of analytes not carrying a phosphate moiety[27]. In general, we cannot exclude that some glycan/peptide combinations still escape detection due to technical reasons (e.g., size, column affinity, ionization) and our report is necessarily biased towards those species we can detect.

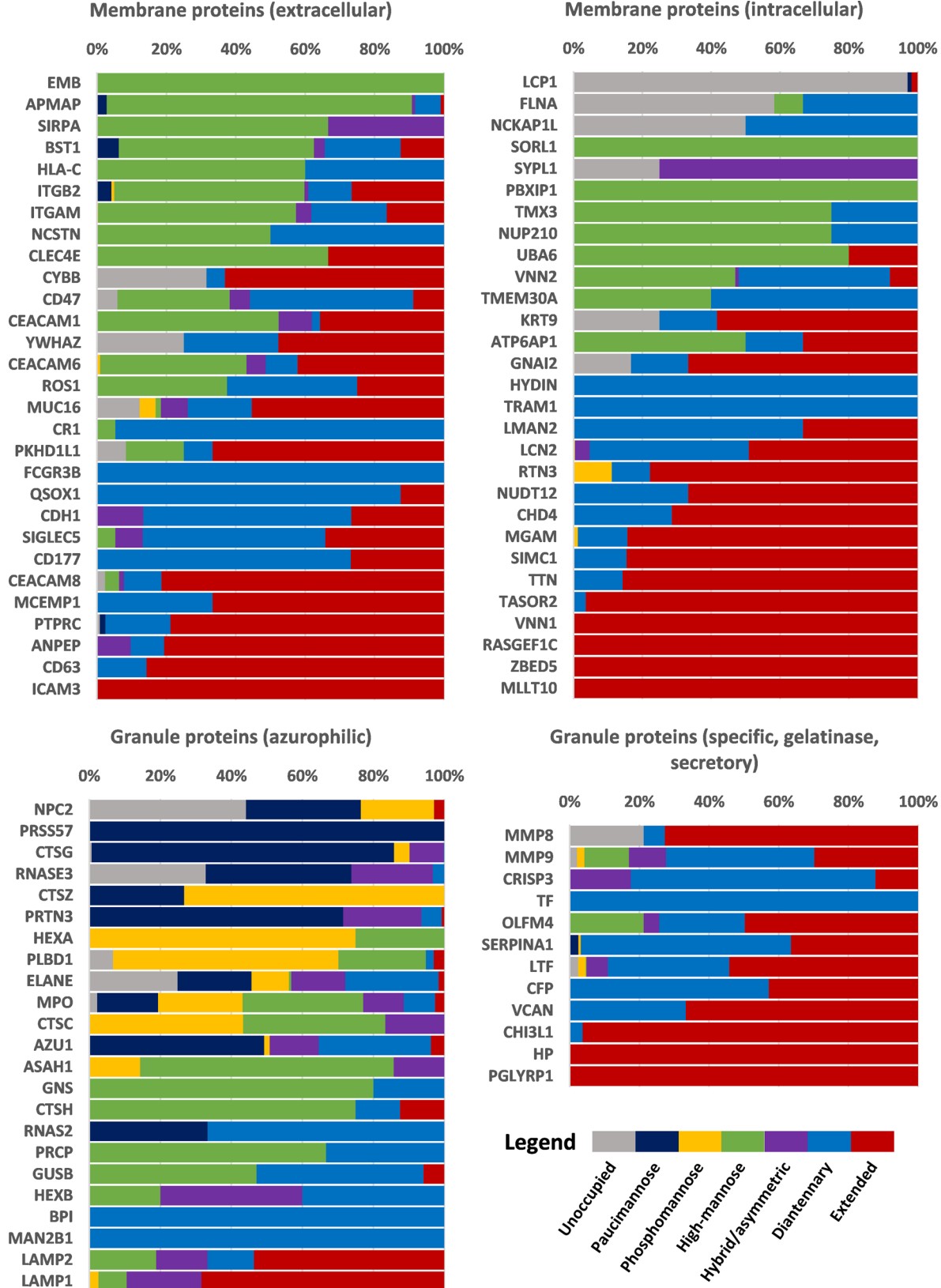

Regarding our assignment of glycan compositions, convincing fragmentation patterns were obtained to prove the presence of both multifucosylated and sialylated glycan species. It has to be noted, however, that the mass increment of sialic acid (+291) is very similar to that of two fucose moieties (+292), and these can

easily be co-isolated for MS$^2$ due to overlapping isotopes. While glycan species with only sialic acids or only fucoses are simple to assign by fragmentation, ratios of sialylation/fucosylation remain challenging to establish from fragment abundances alone. Similarly, we expect our level of sialylation to be somewhat

**Fig. 5 Individual glycoprotein characteristics as a percentage of the total of glycopeptide detections.** Proteins were only included when detected by at least three glycopeptide PSMs (thereby eliminating non-glycosylated proteins). Classification between proteins was made on the cellular location as described in the literature and neXtprot (www.nextprot.org). Both surface and intracellular proteins typically included a transmembrane domain, but this does not necessitate that they always exist membrane-bound. Most intracellular proteins were described as localized to the endoplasmic reticulum or Golgi apparatus, whereas cytosolic, nuclear and mitochondrial proteins typically did not carry N-glycans. While azurophilic (primary) granule proteins are listed separately, the other "granule proteins" included those from the specific (secondary) granules, gelatinase (tertiary) granules and secretory vesicles. For each location, the proteins were sorted based on their glycosylation complexity (following the order of the legend). As evident from the data, the uncommon glycosylation characteristics of pauci- and phosphomannosylation appeared to be defining features of azurophilic granule proteins.

underestimated by the capture of ammonium, or other cations, that dilute their signals and may prevent their observation. Furthermore, our glycosylation type termed "extended" may either be interpreted as tri- or tetraantennary glycosylation or as existing antennae being extended by N-acetyllactosamine (LacNAc) repeats, as previously reported for neutrophil glycosylation[16]. As gaining an antenna is the same mass increment as LacNAc extension, and fragmentation of either typically results in the same oxonium ions (e.g., $m/z$ 366 $[M + H]^+$), we did not make a distinction between them. For the most accurate mass spectrometric quantification of protein glycosylation, we highly recommend isolating single proteins and match the findings from bottom-up MS with intact/native MS as described previously[43].

Lastly, we primarily make use of literature evidence and information available on neXtprot to judge the subcellular localization of glycoproteins[5,29,30], meaning that our proposed link between glycosylation phenotypes and compartmentalization relies on inference rather than direct evidence. In addition, proteins are rarely present in a single compartment alone; azurophilic granules, for instance, are reported to contain 70% of the myeloperoxidase, 84% of the azurocidin, 80% of the myeloblastin, 74% of the cathepsin G, and so forth[5]. Even so, we find it sensible to argue that the dominant glycan distribution of a protein would overlap with its dominant localization, particularly in the case of abundances >50%.

On the putatively annotated membrane-bound proteins (using annotation from neXtprot and specific literature[5,29,30]) both on the cell surface and within the cell, we found a combination of glycosylation sites containing diantennary/extended glycans and those with high-mannose glycans. Individual glycan sites may certainly have unique functions, but it is interesting to see that a large part of the abundant surface receptors is occupied by fucosylated and sialylated glycans. Although our mass spectrometric analysis only yielded monosaccharide compositions and not the linkage between these, it seems likely that the monosaccharides combine into Lewis X (Galβ1,4-[Fucα1,3-]GlcNAc) and sialyl-Lewis X (Neu5Ac2,3-Galβ1,4-[Fucα1,3-]GlcNAc) (Supplementary Fig. 1d), respectively, also named CD15 and CD15s. Particularly CD15 is known to be a constitutive component of neutrophils, both on mature neutrophils as well as on several subpopulations[44,45]. As such, the epitope is generally used as a marker for fluorescence-assisted cell sorting (FACS), including in our study[46,47]. Biologically speaking, sialyl-Lewis X can interact with the E-selectins that are commonly expressed at sites of injury and inflammation. Interactions between E-selectins and neutrophil sialyl-Lewis X motifs may then facilitate the typical tethering behaviour to catch and slow down the cell from the bloodstream and start the extravasation process[46,48].

The proteins likely secreted from specific (secondary) granules, gelatinase (tertiary) granules and other secretory vesicles showed primarily diantennary and extended species as well, but only limited occupancy of high-mannose type glycans. One could argue that high-mannose glycans are therefore a feature of membrane proteins or proteins with a membrane-bound origin. Interestingly, amongst the secretory proteins, we also observed several that have been studied in a variety of biofluids[38,49,50].

Alpha-1-antitrypsin (A1AT), for example, is generally a hepatocyte-produced acute-phase protein that is abundantly present in blood[49]. It is then remarkable to see that our neutrophil-sourced A1AT shows highly similar glycosylation patterns compared to the one from blood, namely highly sialylated (and to a lesser degree fucosylated) di- and triantennary glycans on sites Asn70, Asn107 and Asn271[50,51]. Similarly, lactotransferrin is a major component of human milk and the high degrees of antennary fucosylation we see on both Asn156 and Asn497 and lack of occupancy at Asn642 matches between these highly divergent biofluid sources as well[38]. While it cannot be excluded that, for instance, alpha-1-antitrypsin was co-captured during the flow cytometry of our neutrophils, an alternative explanation is that the glycosylation status of secretory proteins is highly dependent on the structure of the proteins themselves[52].

Most strikingly, the proteins that likely originated from the azurophilic (primary) granules demonstrated quite distinct and typically rare glycosylation characteristics, with sites containing partial unoccupancy, pauci- and phosphomannose glycans, as well as a large degree of hybrid/asymmetric glycans. Not only could this be confirmed for the purified azurophilic granule proteins earlier studied, i.e., myeloperoxidase[24], neutrophil elastase[18], cathepsin G[19], and myeloblastin to some degree[20], but also through our work now expanded to many azurophilic granule proteins including azurocidin, cathepsins C and Z, beta-hexosaminidase subunits alpha and beta, phospholipase B-like 1, and so forth. The glycosidases found amongst the azurophilic granule proteins may also provide a partial explanation for the many paucimannose glycans we observed. Lysosomal alpha-mannosidase (MAN2B1) can cause the removal of the alpha-linked mannoses that occupy most high-mannose and hybrid glycans, whereas the beta-hexosaminidases (HEXA, HEXB) can attack the antennary and core N-acetylglucosamines. If these glycosidases are indeed responsible for the general glycan truncation amongst azurophilic granule proteins, the question instead becomes why certain glycosylation sites remain unaffected. Myeloperoxidase and myeloblastin are intriguing examples of this, myeloperoxidase shows high-mannose glycans on sites Asn355 and Asn391 but severely truncated glycans at Asn729, while myeloblastin shows distinct glycan sizes at Asn129 (HexNAc) and Asn174 ($HexNAc_2Hex_2dHex_1$). Protection of a glycosylation site could be achieved by steric hindrance, strong interactions between the glycan and the peptide backbone (possibly also from another assembly partner), or perhaps the glycans are protected by dedicated chaperones. In all cases, however, the presence of larger glycans in a glycosidase-rich environment reflects an additional level of proteoform regulation.

The other abundantly observed atypical glycosylation characteristic, i.e., phosphomannosylation, may be associated with the trafficking-related function this modification has in other cell types[27,53,54]. Most cells can target glycoproteins to lysosomes, e.g., for degradation, through phosphorylation of high-mannose glycans, which are then subsequently recognized by mannose-6-phosphate receptors and translocated towards the lysosomes[53,54]. While neutrophils do not have typical lysosomes, as they have repurposed them into azurophilic granules (with some distinct

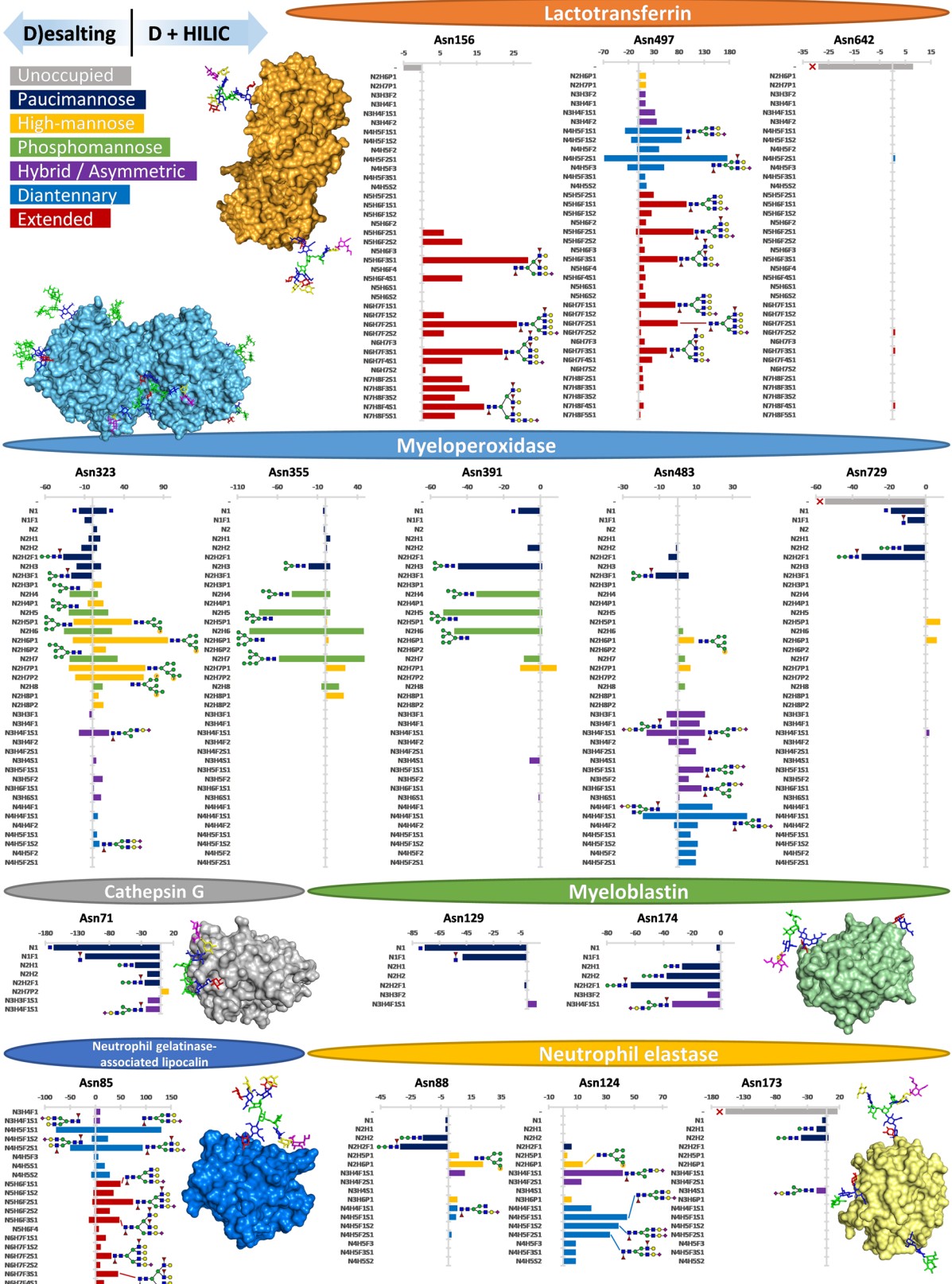

**Fig. 6 Site-specific glycosylation detected on six major neutrophil proteins.** Glycan species were only included in the comparison when at least nine glycopeptide PSMs were detected for any glycosylation site of that protein after either desalting-only (left) or desalting + HILIC (right). Glycoprotein structures were generated by energy-minimization of the glycan on an existing PDB structure by GLYCAM (www.glycam.org). As mass spectrometry does not principally distinguish glycan isomerism, the proposed glycan structures are only estimated on basis of the literature. As can be seen, while proteins exhibited overall glycosylation characteristics such as high complexity (lactotransferrin, neutrophil gelatinase-associated lipocalin) or low complexity (myeloperoxidase, cathepsin G, myeloblastin, neutrophil elastase), they also showed considerable site-dependent heterogeneity (meta-heterogeneity) within the same protein.

differences)[6], it may be that the lysosomal trafficking can still be used to tune granule contents. The dominant theory for populating different granules lies in transcription-based timing, with translated proteins progressively (in time) ending up in azurophilic (primary), specific (secondary) and gelatinase (tertiary) granules[5,55], and this timing has been suggested to influence the glycosylation as well[17]. However, a shuttling mechanism akin to lysosomal trafficking might still allow filling particular granules even when the dedicated time frame has passed. While speculative, this would be an interesting lead to follow-up. Alternatively, both pauci- and phosphomannosylation may assist in the clearance of the hazardous azurophilic granule proteins from the extracellular space. For instance, part of the mannose-6-phosphate receptors can be found on cell surfaces and may mediate the internalization of phosphomannosylated proteins for lysosomal degradation[54,56]. Similarly, other carbohydrate receptors may be instrumental in clearing paucimannose species from the extracellular space. If not cleared, however, both pauci- and phosphomannose glycans could present interesting targets for intervention in neutrophil-based malignancies.

Importantly, we show here that many of the neutrophil proteins commonly targeted by anti-neutrophil cytoplasmic antibodies (ANCAs), autoantibodies that are found in a variety of autoimmune diseases, are also those that are (partially) occupied by pauci-mannose and phosphomannose glycans, next to exhibiting site-dependent macro-heterogeneity. Examples of these proteins include myeloperoxidase, myeloblastin (proteinase 3), lactotransferrin, neutrophil elastase, and cathepsin G, each of these displaying the uncommon glycosylation characteristics[9–11]. One can reason that progressively truncated glycans may lead to exposure (or development) of antigenic protein regions, destabilization of protein structure, or that the uncommon glycans themselves already present an immunogenic epitope. Conversely, the shared glycosylation characteristics of the ANCA targets may also provide an interesting opportunity for diagnosis and/or treatment in autoimmune disease, as targeted methods might be developed that focus on pauci- or phosphomannose residues. While the link between autoimmunity and uncommonly glycosylated ANCA targets needs substantial further investigation, we believe that our study has highlighted important shared characteristics amongst many of the involved proteins.

To conclude, we present here a comprehensive investigation of the neutrophil glycoproteome and report on the site-specific glycosylation of more than 200 individual neutrophil glycoproteins. Next to characterizing the glycopeptides of proteins that are generally membrane-bound and secreted, we report the distinctive glycan characteristics for proteins that typically reside in the azurophilic granules. As many of these proteins are also commonly targeted by antibodies in autoimmune diseases, we believe our investigation has described exciting potential leads into the mechanisms and future treatment of neutrophil-based malignancies.

## Methods
**Chemicals and materials**. Unless otherwise specified, all chemicals and reagents were obtained from Sigma-Aldrich (Steinheim, Germany). GluC was obtained from Roche (Indianapolis, IN). The Oasis PRiME HLB plate was purchased from Waters (Etten-Leur, the Netherlands). Formic acid (FA) was purchased from Merck (Darmstadt, Germany). Acetonitrile (ACN) was purchased from Biosolve (Valkenswaard, The Netherlands). Milli-Q was produced by an in-house system (Millipore, Billerica, MA), phosphoSTOP (Roche, Woerden, the Netherlands) and complete mini EDTA free (Roche, Woerden, the Netherlands).

**Isolation of human neutrophils**. Buffy coats were obtained from Sanquin's blood bank. For this, blood from 10 healthy donors was mixed and neutrophils were enriched by Percoll (GE Healthcare, Sweden) density gradient centrifugation for 20 min at 2000×g. The supernatant was removed until 0.5 cm above the red pellet, the pellet was mixed with erythrocyte lysis buffer containing NH₄Cl (Sigma

Aldrich, Germany), KHCO₃ (Merck, Germany) and EDTA (Amresco, USA) and kept on ice until the colour changed from bright red to dark red. Cells were centrifuged for 5 min at 500×g and washed once with PBS. $10^8$ Cells were labelled with CD15 BB515), CD45 BV421 (both BD Biosciences, USA) and CD157 PE (eBiosciences, USA) for 15 min at room temperature (RT), washed once with PBS. Neutrophils were sorted on a FACS ARIA III flow cytometer (BD Biosciences, USA) as having high side scatter and intermediate CD45 expression and high CD15 and CD157 expression with dim autofluorescence in the AF430 channel. The purity of the sorted neutrophils was determined to be >98% (Supplementary Fig. 4). The cells were finally collected in 15 mL tubes, washed once with PBS, and snap-frozen as a dry pellet in liquid nitrogen before storage at −80 °C.

**Cell lysis and protein digestion**. Neutrophil cell pellets (~$3 \times 10^7$ cells) were resuspended in lysis buffer containing 100 mM Tris–HCl (pH 8.5), 7 M urea, 5 mM Tris(2-carboxyethyl)phosphine hydrochloride (TCEP), 30 mM chloroacetamide (CAA), Triton X-100 (1%), 2 mM magnesium sulfate, phosphoSTOP and complete mini EDTA free. Then, cells were disrupted by sonication for 45 min (alternating 20 s on and 40 s off) using a Bioruptor Plus (Diagenode, Seraing, Belgium). Cell debris was removed by centrifugation at 14,000 rpm for 1 h at 4 °C and the supernatant was retained. Impurities were removed by methanol/chloroform protein precipitation as follows: 1 mL of supernatant was mixed with 4 mL of methanol, 1 mL chloroform and 3 mL ultrapure water with thorough vortexing after each addition. The mixture was then centrifuged for 10 min at 5000 rpm at RT. The upper layer was discarded, and 3 mL of methanol was added. After sonication and centrifugation (5000 rpm, 10 min at RT), the solvent was removed, and the precipitate was allowed to dry by air. The pellet was resuspended in digestion buffer containing 100 mM Tris–HCl (pH 8.5), 1% w/v sodium deoxycholate (SDC), 5 mM TCEP and 30 mM CAA. GluC was then added to digest proteins for 3 h at an enzyme-to-protein ratio of 1:75 (w/w) at 37 °C, and the resulting peptide mixtures were further digested overnight at 37 °C by trypsin (1:20; w/w). The next day, SDC was removed via acid precipitation (0.5% trifluoroacetic acid) (TFA) and the final peptide concentration was estimated by measuring the absorbance at 280 nm on a Nanodrop spectrophotometer (Nanodrop 2000, Thermo Scientific). The peptides were desalted by using an Oasis PRiME HLB plate then dried and stored at −80 °C.

**HILIC-based glycopeptide enrichment**. The HILIC-based glycopeptide enrichment was performed using in-house packed stage-tips. In short, 200 μL pipet tips were packed with 10 mg of 3 μm 100 Å ZIC-cHILIC beads (Merck, Darmstadt, Germany), which were then washed with 100 μL of 1% FA and equilibrated with 100 μL of loading buffer (80% ACN/0.5% TFA). The peptide digests (100 μg) were reconstituted with 100 μL of loading buffer and loaded onto the stage-tips. The stage-tips were washed with 100 μL of loading buffer, and the glycopeptides were first eluted with 65% ACN/0.5% TFA, followed by 55% ACN/0.5% TFA. The elution was dried down and stored at −80 °C until subjected to LC–MS².

**Shotgun proteomics**. Shotgun LC–MS² was performed by means of an Agilent 1290 Infinity HPLC system (Agilent Technologies, Waldbronn, Germany) coupled to a Q Exactive HF mass spectrometer (Thermo Fisher Scientific, Bremen, Germany). Per run, ~800 ng of peptides were first trapped by using a 100 μm inner diameter 2 cm trap column (in-house packed with ReproSil-Pur C18-AQ, 3 μm) (Dr. Maisch GmbH, Ammerbuch-Entringen, Germany) coupled to a 50 μm inner diameter 50 cm analytical column (in-house packed with Poroshell 120 EC-C18, 2.7 μm) (Agilent Technologies, Amstelveen, The Netherlands). The mobile-phase solvent A consisted of 0.1% FA in water, and the mobile-phase solvent B consisted of 0.1% FA in ACN. Trapping was performed at a flow rate of 5 μL/min for 5 min with 0% B and peptides were eluted using a passively split flow of 300 nL/min for 170 min with 10–36% B over 155 min, 36–100% B over 3 min, 100% B for 1 min, 100–0% B over 1 min, and finally held at 0% B for 10 min. Peptides were ionized using a spray voltage of 1.9 kV and a heated capillary. The mass spectrometer was set to acquire full-scan MS spectra (m/z 375–1600) for a maximum injection time of 20 ms at a mass resolution of 60,000 and an automated gain control (AGC) target value of $3 \times 10^6$. Up to 15 of the most intense precursor ions were selected for tandem mass spectrometry (MS²). HCD MS² (m/z 200–2000) acquisition was performed in the HCD cell, with the readout in the Orbitrap mass analyser at a resolution of 30,000 (isolation window of 1.4 Th) and an AGC target value of 1e5 or a maximum injection time of 50 ms with a normalized collision energy of 27%.

**Glycoproteomics**. All peptides from the (glyco)peptide enrichments were separated and analysed using the same HPLC system as used for the global proteome analysis, albeit now coupled on-line to an Orbitrap Fusion Lumos mass spectrometer (Thermo Fisher Scientific, Bremen, Germany) using a 90 min gradient, as follows: 0–5 min, 100% solvent A; 13–44% solvent B for 65 min; 44–100% solvent B for 5 min; 100% solvent B for 5 min; 100% solvent A for 15 min. Per run, ~300 ng (nonenriched) or 100 μg (HILIC enriched) peptides were ionized using a 2.0 kV spray voltage. For the MS scan, the mass range was set from m/z 350–2000 with a maximum injection time of 50 ms at a mass resolution of 60,000 and an AGC target value of $5 \times 10^4$ in the Orbitrap mass analyser. The dynamic exclusion was

set to 30 s for an exclusion window of 10 ppm with a cycle time of 3 s. Charge-states screening was enabled, and precursors with 2+ to 8+ charge states and intensities > 1e5 were selected for MS$^2$. HCD MS$^2$ (*m/z* 120–4000) acquisition was performed in the HCD cell, with the readout in the Orbitrap mass analyser at a resolution of 30,000 (isolation window of 2 Th) and an AGC target value of $5 \times 10^4$ or a maximum injection time of 50 ms with a normalized collision energy of 27%. If at least 2 out of 4 oxonium ions of glycopeptides (138.0545+, 204.0687+, 366.1396+ or 243.026+) were observed[28], EThcD MS$^2$ on the same precursor was triggered (isolation window of 3 Th) and fragment ions (*m/z* 120–4000) were analysed in the Orbitrap mass analyser at a resolution of 30,000, AGC target value of $2 \times 10^5$ or a maximum injection time of 250 ms with activation of ETD and supplemental activation with normalized collision energy (NCE) of 27%.

**Proteomics data analysis**. Proteome Discoverer (version 2.3, Thermo Scientific) was used to analyse the LC–MS$^2$ raw files from the shotgun experiment. MS$^2$ scans were searched against the Swiss-Prot database (release date: September 2019, 20,421 entries, taxonomy: *Homo sapiens*) by using Sequest with the following modifications: fixed Cys carbamidomethylation, variable Met oxidation and Ser, Thr, and Tyr phosphorylation (serine, threonine and tyrosine). Enzyme specificity was set to trypsin and GluC with a maximum of three missed cleavages for peptides in fully specific digestion and set to "none" for peptides in nonspecific digestion. The searches were performed using a precursor mass tolerance of 10 ppm and a fragment mass tolerance of 0.05 Da followed by data filtering using Percolator, resulting in a 1% false discovery rate (FDR). For label-free quantification, the node called Minora feature detector was used with high PSM confidence, a minimum of 5 non-zero points in a chromatographic trace, a minimum number of 2 isotopes and a maximum retention time difference of 0.2 min for isotope peaks. The consensus workflow in Proteome Discoverer was used to open the search results and enable retention time (RT) alignment with a maximum RT shift of 5 min and mass tolerance of 10 ppm in order to match the precursor between runs. Proteins were only reported when detected in at least two out of three replicates and covered by at least three unique peptides.

**Glycoproteomics data analysis**. The raw files acquired from the glycoproteomics experiments were searched with Byonic (version 3.7.13, Protein Metrics Inc.)[33] against all potential glycoproteins (sequences containing Asn-Xxx-Ser/Thr where Xxx ≠ Pro) from the Swiss-Prot database (one replicate) as well as against a focused database comprising the top 500 most abundant glycoproteins from the neutrophil proteome analysis (all replicates). The precursor ion mass tolerance was set to 10 ppm and both HCD & EThcD fragment mass tolerance were set to 20 ppm. Peptides were searched with nonspecific proteolytic digestion, Cys carbamidomethylation as fixed modification, and up to two variable modifications of Met oxidation and Ser/Thr/His/Tyr phosphorylation. For glycosylation, we allowed one glycan from a database of 279, which included highly fucosylated species and phosphomannose/degraded glycans known from the lysosomal pathway of degradation (Supplementary Data 1). Based on prior established curation criteria[35], we rejected glycopeptide identifications with a log probability lower than 1 and maintained a score threshold of 150. These criteria retained 85 reverse (glyco)peptide PSMs out of a total of 15,095, establishing the FDR to be <1%. PSMs of individual glycan compositions were counted using an in-house script and were either displayed directly to retain the relative data density between enrichment methods (Fig. 6), or were normalized to the sum of counts per enrichment to highlight the respective distribution of glycoforms (Fig. 4).

**Data visualization**. For visualization of glycan species, we followed the recommendations of the Consortium for Functional Glycomics[57], while glycan cartoons were constructed using Glycoworkbench 2.1 (build 146)[58]. Glycoprotein structures were generated using GLYCAM[59] using the following PDB structures: lactotransferrin (1N76), myeloperoxidase (1CXP), cathepsin G (1AU8), myeloblastin (one chain of 1FUJ), neutrophil gelatinase-associated lipocalin (1QQS), and neutrophil elastase (one chain of 3Q76). Glycan compositions were selected to be an abundant and representative species based on the glycoproteomics data, while linkages thereof were only estimated and may differ in actuality.

**Reporting summary**. Further information on research design is available in the Nature Research Reporting Summary linked to this article.

## Data availability

All raw data and processed files generated throughout the study are available through the MassIVE repository accessible via ftp://MSV000086020@massive.ucsd.edu using the password "NPGP_01". Any remaining information can be obtained from the corresponding author upon reasonable request.

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

## Acknowledgements

This work was supported by the Netherlands Organization for Scientific Research (NWO) TOP-Punt Grant 718.015.003 (to A.J.R.H.) and the European Union's Horizon 2020 Research and Innovation Programme under Grant 668036 (RELENT). We further acknowledge funding for our large-scale proteomics facility, the Netherlands Proteomics Center, through the X-omics Road Map programme (project 184.034.019) and the EU Horizon 2020 programme INFRAIA project Epic-XS (Project 823839). K.R.R. acknowledges additional support from NWO Veni project VI.Veni.192.058. Not least, we thank Tomislav Čaval for critically reading the manuscript.

## Author contributions

K.R.R., Y.-H.L. and A.J.R.H. designed the research and planned the experiments. F.P.J.A. and A.B.M. collected the neutrophils and performed FACS. K.R.R. and Y.-H.L. performed the proteomics and glycoproteomics experiments as well as the data analysis thereof. All authors contributed to the writing of the manuscript.

## Competing interests

The authors declares no competing interests.
