## [Transparent Peer Review File · Communications Biology]

Reviewers' comments:

Reviewer #1 (Remarks to the Author):

The authors performed conventional proteomics and glycoproteomics on protein extracts from lysates of pooled neutrophils obtained from ten healthy donors. The site-specific glycoprofile of approximately 200 proteins were reported. Information of the relative protein abundance, site occupancy and glycan composition but not structure was achieved. As expected, glycosylation features previously reported for neutrophils were suggested from the glycan compositional information. The subcellular location (granule origins, membrane/soluble etc) and biological functions (antigenic, proteolytic, microbicidal etc) of the identified glycoproteins were inferred from literature, and discussed at length without empirical evidence or support from other experimental approaches. The site-specific glycosylation (rather than "meta-heterogeneity", I do not agree that a new word is required to describe this phenomenon) of key neutrophil proteins was discussed from the obtained data (e.g. Fig 6), several of which have already been characterized in greater details by others. More critically, the findings and conclusions presented in this study are not as novel as claimed by the authors. Acknowledgement of key literature is missing. Human neutrophils have repeatedly been studied with transcriptomics, (glyco)proteomics and glycomics some with a more informative functional and disease angle and some with subcellular fractionation. To this end, I do not find that this study significantly advances our understanding of neutrophil biology. The work may with some improvements be better suited as a technical note in a specialist journal.

Reviewer #2 (Remarks to the Author):

This is a rather well-accomplished report from a well-established lab. The glycoproteomic workflow adopted for the analysis of neutrophil is sound and offers a global view of the glycan distribution among the more abundant glycoproteins identified. It shows that pauci and phosphomannose structures are particularly abundant on glycoproteins from azurophilic granules. This characteristic could be contrasted against the membrane and secreted glycoproteins that carry more of the high mannose and complex type structures, including epitopes inferred to be LeX. It provides a good reference for those interested or working on the neutrophil biology albeit not without its shortcomings typical of all current glycoproteomic studies. Most technical issues were discussed or touched upon and there are no significant problems found. However, readers would benefit from a few clarifications that should be provided, as noted below.

1. Figure 1 is informative. However, it seems that the protein ID and LFQ in the shotgun proteomics part did not use MaxQuant as indicated, based on the description provided in the Methods.
2. The desalted peptides/glycopeptides were analyzed either without further enrichment or by HILIC. Considering the authors have previously developed Fe³⁺-IMAC-based methods for phosphomannosylated glycopeptides and since this class of glycans feature prominently in neutrophil and this report, the authors should comment on whether a better coverage may benefit from such additional specialized enrichment.
3. Although the usual threshold for confident Byonic ID has been applied, most of the b and y ions used to assign sequence remain visibly weak (as demonstrated in Fig 3). No exemplary ETHcD spectrum was shown to demonstrate how it may have helped assignment. In view of searching against a wide range of glycan possibilities and non-specific protease cleavages, with the common issues of correct monoisotopic precursor assignment, ambiguity among different permutation of glycosyl compositions vs peptide modifications, mis-cleavages etc, manual verification would be necessary and yet tedious and also not always foolproof either. More clarification with respect to the integrity of the dataset and the potential level of mis-assignment will be helpful to uninitiated readers.

4. The authors were careful not to include any sialyl LeX epitope in Fig 4 for glycans carrying multiple Fuc and NeuAc. Can the oxonium ion for NeuAc1Fuc1Hex1HexNAc1 be detected either by HCD or EThcD in any of the PSM? Would varying the collision energy used be of any use? It seems that the same normalized CE was used for HCD and EThcD.

5. "PSMs of individual glycan compositions were counted using an in-house script and relatively quantified by total area normalisation of the sum of counts, either per glycosylation site (Figure 6), per glycoprotein (Figure 6), or across the whole glycoproteome (Figure 4)." - This should be further elaborated. The scale used for Fig 4 is in the range of ± 0.01 , whereas that for Fig 6 can be ± 180 . What these numbers represent? How to counter the problem that non-detection does not mean non-existence? While positive detection of various glycans on the glycopeptides can be used to calculate the glycoform distribution, how to determine the relative amount of non-occupied site? by the PSM count for the peptide with non-occupied Asn? How would these translate into % unoccupied for glycoproteins containing more than one sites in Fig 5?

Reviewer #3 (Remarks to the Author):

Site-specific glycosylation analysis of complex samples is extremely difficult and challenging. Neutrophils have phagocytosis and digestion functions and are an important component of the innate immune system. Neutrophils are the body's first line of defense against pathogenic microorganisms and can also regulate innate and adaptive immunity, mediating anti- and pro-tumor activities as well as autoimmune disease. In this manuscript, Reiding et al. decoded the glycoproteome and their site-specific glycosylation in neutrophils. Intriguingly, the authors analyzed and compared the glycan occupancy on membrane and secreted glycoproteins, which is helpful to reveal the relationship between neutrophil glycoproteins and their function. The manuscript is generally well written and can be published in *Communications Biology* after minor revisions.

1. It is better for the author to discuss that why they did not benchmark label-free quantification for non-tryptic peptides (Line 118).

2. In proteome data analysis, the searches were performed using a fragment mass tolerance of 0.05 Da (Line 451), which is too loose for protein identification.

3. The authors found some potential glycosylation sites, such as Asn642 and Asn173, were mostly unoccupied. Whether these enzymatic peptides are too long and are difficult for ionization in the presence of occupied glycans needs to be analyzed and discussed.

Merry Christmas and Happy New Year !

Response to reviewers' comments:

Reviewer #1 (Remarks to the Author):

Reviewer #1: The authors performed conventional proteomics and glycoproteomics on protein extracts from lysates of pooled neutrophils obtained from ten healthy donors. The site-specific glycoprofile of approximately 200 proteins were reported. Information of the relative protein abundance, site occupancy and glycan composition but not structure was achieved. As expected, glycosylation features previously reported for neutrophils were suggested from the glycan compositional information.

Answer: We thank the reviewer for their time and effort spent on critically reading our manuscript. Indeed, we primarily report compositional information, with some structural components determinable by fragmentation analysis. We were happy to find that several glycan features did overlap with previous reports, while our work expands on this with site-specific glycosylation information on a much large number of glycoproteins. Important new findings from our investigation are the presence of phosphomannosylated glycans across a subset of neutrophil proteins, as well as the determination of which major proteins carry paucimannose species.

Reviewer #1: The subcellular location (granule origins, membrane/soluble etc) and biological functions (antigenic, proteolytic, microbicidal etc) of the identified glycoproteins were inferred from literature, and discussed at length without empirical evidence or support from other experimental approaches.

Answer: We find it important to reflect on our findings and to make use of established knowledge where possible and appropriate. For example, compartmentalization of neutrophil proteins has been well-described (**Rorvig S., et al., 2013, J Leukoc Biol**; **Grabowski P., et al., 2019, Mol Cell Proteomics**; **Rieckmann, J.C., et al., 2017, Nat Immunol**) and we find this knowledge valuable to categorize our findings and place them in biological context. Now that our analytical methodology for neutrophil glycoproteomics has been established, we will pursue functional experiments in the near future.

Reviewer #1: The site-specific glycosylation (rather than “meta-heterogeneity”, I do not agree that a new word is required to describe this phenomenon) of key neutrophil proteins was discussed from the obtained data (e.g. Fig 6), several of which have already been characterized in greater details by others.

Answer: Glycoproteomics approaches on “single” proteins will likely always be more in-depth than global analyses, as demonstrated by others and ourselves (**Loke I., et al., 2017, Mol Cell Proteomics**; **Loke I., et al., 2015**; **Reiding K.R., et al., 2019, J Biol Chem**). However, very few studies demonstrate whole-cell glycoproteomics, as we perform here, to uncover distinctly glycosylated protein subsets. For the proteins that overlap between the detailed analyses and our whole-cell study, *e.g.*, neutrophil elastase, cathepsin G and myeloperoxidase, we are happy to report highly similar findings that validate our other observations, and make we believe fair references to that earlier work.

We encourage using both the concepts of site-specific glycosylation and meta-heterogeneity, but in a different context. We consider site-specific glycosylation to be the glycosylation on a given site (describable in terms of macro- and micro-heterogeneity), but meta-heterogeneity to be the distribution of site-specific glycosylations across a whole protein. While everyone is of course free to use his/her preferred concepts, we have discussed our rationale for using the term meta-heterogeneity earlier in **Caval T., et al., 2020, Mol Cell Proteomics**.

Reviewer #1: More critically, the findings and conclusions presented in this study are not as novel as claimed by the authors. Acknowledgement of key literature is missing. Human neutrophils have repeatedly been studied with transcriptomics, (glyco)proteomics and glycomics some with a more informative functional and disease angle and some with subcellular fractionation. To this end, I do not find that this study significantly advances our understanding of neutrophil biology. The work may with some improvements be better suited as a technical note in a specialist journal.

Answer: We do not fully understand this point, as we have tried to be inclusive with citing the literature, with reports on individual proteins (**Loke I., et al., 2017, Mol Cell Proteomics; Loke I., et al., 2015; Reiding K.R., et al., 2019, J Biol Chem**), transcriptomics (**Rorvig S., et al., 2013, J Leukoc Biol**) as well as previous proteomics and glycoproteomics approaches (**Thaysen-Andersen M., et al., 2015, J Biol Chem; Grabowski P., et al., 2019, Mol Cell Proteomics; Rieckmann, J.C., et al., 2017, Nat Immunol**).

We do consider a recent publication to be relevant to our story as well, namely **Venkatakrishnan V., 2020, J Biol Chem**, but this has only appeared after the initial submission of our manuscript. We have now included the reference together with the following text in the introduction:

“Recent glycomics (released glycan) evidence points towards the azurophilic granules to contain at least paucimannosylation via a transcription-based timing mechanism, but phosphomannose glycans remain challenging to localize and the involved protein carriers remain to be elucidated [Venkatakrishnan V., 2020, J Biol Chem].”

In all, we feel confident that our detection of phospho- and paucimannose glycosylation across the neutrophil proteome, but primarily on proteins known to reside in the azurophilic granules, is a new finding and of considerable biological interest. We have yet to find a neutrophil study to reach the same conclusions.

Reviewer #2 (Remarks to the Author):

Reviewer #2: This is a rather well-accomplished report from a well-established lab. The glycoproteomic workflow adopted for the analysis of neutrophil is sound and offers a global view of the glycan distribution among the more abundant glycoproteins identified. It shows that pauci and phosphomannose structures are particularly abundant on glycoproteins from azurophilic granules. This characteristic could be contrasted against the membrane and secreted glycoproteins that carry more of the high mannose and complex type structures, including epitopes inferred to be LeX. It provides a good reference for those interested or working on the neutrophil biology albeit not without its shortcomings typical of all current glycoproteomic studies. Most technical issues were discussed or touched upon and there are no significant problems found. However, readers would benefit from a few clarifications that should be provided, as noted below.

Answer: We appreciate the reviewer’s supportive words, as well as the recommendations to improve the clarity of our manuscript.

Reviewer #2: 1. Figure 1 is informative. However, it seems that the protein ID and LFQ in the shotgun proteomics part did not use MaxQuant as indicated, based on the description provided in the Methods.

Answer: We thank the reviewer for noticing and have adjusted Figure 1 to reflect our use of Proteome Discoverer rather than MaxQuant.

Reviewer #2: 2. The desalted peptides/glycopeptides were analyzed either without further enrichment or by HILIC. Considering the authors have previously developed Fe³⁺-IMAC-based methods for phosphomannosylated glycopeptides and since this class of glycans feature prominently in neutrophil and this report, the authors should comment on whether a better coverage may benefit from such additional specialized enrichment.

Answer: We definitely expect Fe³⁺-IMAC to increase the coverage of phosphomannosylated glycopeptides, as reported in **Caval T., 2019, Mol Cell Proteomics**. However, such an approach would limit the comparability between phosphomannosylated glycoproteins and their non-phosphorylated counterparts. We find this an interesting point and have added our thoughts to the discussion:

“Other enrichment methods could still prove valuable for exploring different parts of the neutrophil glycoproteome. For example, analysis of phosphomannosylated glycoproteins and glycopeptides could be improved by making use of Fe³⁺-IMAC enrichment, although this would be at the expense of analytes not carrying a phosphate moiety [Caval T., 2019, Mol Cell Proteomics].”

Reviewer #2: 3. Although the usual threshold for confident Byonic ID has been applied, most of the b and y ions used to assign sequence remain visibly weak (as demonstrated in Fig 3). No exemplary ETHcD spectrum was shown to demonstrate how it may have helped assignment. In view of searching against a wide range of glycan possibilities and non-specific protease cleavages, with the common issues of correct monoisotopic precursor assignment, ambiguity among different permutation of glycosyl compositions vs peptide modifications, mis-cleavages etc, manual verification would be necessary and yet tedious and also not always foolproof either. More clarification with respect to the integrity of the dataset and the potential level of mis-assignment will be helpful to uninitiated readers.

Answer: We agree with the reviewer that exemplary ETHcD spectra would have been a sensible inclusion and we have now added the following figure to the Supporting Information:

“Figure S1: Neutrophil glycopeptide MS/MS spectra as generated by EThcD. A) Localization of the GlcNAc to myeloblastin Asn129, most distinguishable by clear C8 and C9 fragments carrying the monosaccharide residue. B) Localization of paucimannose glycosylation to myeloblastin Asn174, indicated by glycosylated C2 and C3 fragments. C) Behaviour of phosphomannosylated neutrophil elastase glycopeptides in EThcD. D) Lactoferrin glycopeptide showing HexNAc₁Hex₁Fuc₁NeuAc₁ oxonium ions (m/z 657.23), suggesting sialyl-Lewis A/X to be at least partially present for this glycopeptide composition.”

Ambiguity and misassignments are important considerations for (glyco)proteomics data. While we have indeed performed various steps of curation throughout our data analysis, as detailed in the Methods section, the uninitiated reader might benefit most from a false-discovery rate established on a dummy

(reversed) database. Throughout the MS runs, the curation retained 85 reverse (glyco)peptide PSMs out of a total of 15095 (= 0.56%, or <1%). We have added this information to the revised manuscript.

"... These criteria retained 85 reverse (glyco)peptide PSMs out of a total of 15095, establishing the FDR to be <1%."

Reviewer #2: 4. The authors were careful not to include any sialyl LeX epitope in Fig 4 for glycans carrying multiple Fuc and NeuAc. Can the oxonium ion for NeuAc₁Fuc₁Hex₁HexNAc₁ be detected either by HCD or ETHcD in any of the PSM? Would varying the collision energy used be of any use? It seems that the same normalized CE was used for HCD and ETHcD.

Answer: Variation in collision energy, possibly in the same run in the form of stepping-energy HCD, would have been a good alternative to ETHcD in terms of sequencing both the glycan and the peptide. As it is, we have selected panel D of the new Supporting Information figure (see previous comment) to display a glycopeptide with a distinguishable HexNAc₁Hex₁Fuc₁NeuAc₁ fragment, typically indicative of sialyl-Lewis X. However, this recording likely still comprises a blend of structural isomers, while fucoses are known to migrate during mass spectrometry. As such, we would like to be conservative in assigning sialyl-Lewis X across our measurements (for now).

Reviewer #2: 5. "PSMs of individual glycan compositions were counted using an in-house script and relatively quantified by total area normalisation of the sum of counts, either per glycosylation site (Figure 6), per glycoprotein (Figure 6), or across the whole glycoproteome (Figure 4)." - This should be further elaborated. The scale used for Fig 4 is in the range of ± 0.01 , whereas that for Fig 6 can be ± 180 . What these numbers represent?

Answer: In both cases these numbers represent PSMs, but Figure 4 shows the percentage across detections, while Figure 6 just shows detections. For Figure 4 we found it important to normalize the scales so that the detection heterogeneity (small glycans vs. big glycans) can be better appreciated. For Figure 6, on the other hand, we found it important to highlight the relative data density (few PSMs vs. many PSMs) so that a better estimate can be made of the occupancy per individual site. We agree that our description of this is unclear and have altered the method section:

"PSMs of individual glycan compositions were counted using an in-house script and were either displayed directly to retain the relative data density between enrichment methods (Figure 6), or were normalized to the sum of counts per enrichment to highlight the respective distribution of glycoforms (Figure 4)."

Reviewer #2: How to counter the problem that non-detection does not mean non-existence? While positive detection of various glycans on the glycopeptides can be used to calculate the glycoform distribution, how to determine the relative amount of non-occupied site? by the PSM count for the peptide with non-occupied Asn? How would these translate into % unoccupied for glycoproteins containing more than one sites in Fig 5?

Answer: We have indeed used peptides without glycan (but containing the N-glycan sequon) to quantify non-occupancy. While many factors are likely to bias this quantification (see also our response to Reviewer #3), we do expect to detect this non-glycosylated peptide more often if the site is only partially occupied. Figure 5 is a simplification in this regard, as it groups all sites into one graph. However, we have included all MS data in the Supporting Information, and we are happy to visualize site-specific glycosylation (like Figure 6) for other proteins as well should such a request occur.

Non-detection is indeed very damaging for quantification, but also very difficult to prevent. To obtain the best overview of an individual protein's glycoproteoforms we highly recommend its analysis by intact or native MS, but this is still challenging for complex protein mixtures.

Reviewer #3 (Remarks to the Author):

Reviewer #3: Site-specific glycosylation analysis of complex samples is extremely difficult and challenging. Neutrophils have phagocytosis and digestion functions and are an important component of the innate immune system. Neutrophils are the body's first line of defense against pathogenic microorganisms and can also regulate innate and adaptive immunity, mediating anti- and pro-tumor activities as well as autoimmune disease. In this manuscript, Reiding et al. decoded the glycoproteome and their site-specific glycosylation in neutrophils. Intriguingly, the authors analyzed and compared the glycan occupancy on membrane and secreted glycoproteins, which is helpful to reveal the relationship between neutrophil glycoproteins and their function. The manuscript is generally well written and can be published in Communications Biology after minor revisions.

Answer: We thank the reviewer for their time, effort and supportive words, and appreciate the comments to improve our manuscript.

Reviewer #3: 1.It is better for the author to discuss that why they did not benchmark label-free quantification for non-tryptic peptides (Line 118).

Answer: It is our estimation that many critical factors remain to be investigated to achieve label-free quantification of non-tryptic peptides. These include protease specificities, charge affinity and distribution, fragmentation propensity and even factors like database search biases (many algorithms are trained on lysine-terminating peptides). While critical to move the field of proteomics forward, we do feel this investigation is beyond the scope of the here-presented neutrophil glycoproteomics investigation. Nevertheless, we have changed line 118 to the following to reflect these considerations:

“While we did not benchmark label-free quantification for non-tryptic peptides - differences in charge affinity and distribution, fragmentation propensity and interpretability by search engines each demand further investigation - this substantial enlargement of the detectable proteome by using a non-specific search strategy indicates that endogenous processing should be accounted for when analysing neutrophils.”

Reviewer #3: 2.In proteome data analysis, the searches were performed using a fragment mass tolerance of 0.05 Da (Line 451), which is too loose for protein identification.

Answer: For our proteomics searches we made use of settings that were equivalent to previous publications with samples of similar complexity (Zhu J, 2019, J Proteome Res; Zhu J, 2020, J Proteome Res), which encompassed the value for the fragment mass tolerance. While we believe there are certainly benefits to a fragment mass tolerance that is stricter than 0.05 Da (e.g., allowing for a less strict post-processing curation), our current analysis workflow controls the false discovery rate to be 1% at best for both the peptide and protein levels. While this means that false detections will inevitably remain present in our dataset, the frequency thereof should be very limited.

Reviewer #3: 3.The authors found some potential glycosylation sites, such as Asn642 and Asn173, were mostly unoccupied. Whether these enzymatic peptides are too long and are difficult for ionization in the presence of occupied glycans needs to be analyzed and discussed.

Answer: We do agree that large glycans may complicate ionization and detection of glycopeptides, but especially after HILIC enrichment we are able to detect large glycan compositions across a variety of peptides (tetra-antennary and larger, **Figure 4**). The unoccupancy of lactoferrin Asn642 has prior been established by methods other than mass spectrometry (**van Berkel P.H., et al., 1996, Biochem J**) and we are happy to find similar data. For partially occupied sites like myeloperoxidase Asn729 and elastase Asn173 we also detect dozens of PSMs that are glycosylated, so in these cases we do have a reasonable frame of reference to quantify against. In general, however, we cannot exclude that some glycan/peptide combinations escape our detection for technical reasons and that results are biased towards those species that we can detect (including non-occupancy). We have included the above reference in the manuscript and have added a reflection on the matter in the Discussion:

“Lactotransferrin Asn642 appeared mostly unoccupied, matching previous observations that did not rely on MS for detection [van Berkel P.H., et al., 1996, Biochem J]”

“In general, we cannot exclude that some glycan/peptide combinations still escape detection due to technical reasons (e.g., size, column affinity, ionization) and our report is necessarily biased towards those species we can detect.”

Reviewer #3: Merry Christmas and Happy New Year !

Answer: While time is ever-moving, we do wish all reviewers and editors a great year as well.

Reviewers' comments:

Reviewer #1 (Remarks to the Author):

The authors have submitted a lightly revised R1 manuscript. After another detailed read, it is still my view that this manuscript, which applies a single line of enquiry and which relies on inference rather than direct evidence to reach biological conclusions (many of which have been shown before with a greater level of evidence) is better suited in a lower ranking specialist journal. Moreover, multiple critical issues were identified as detailed below. Finally, to put the study and its findings in a perspective, I think it deserves to be mentioned that a conventional glycoproteomics experiment as performed with this work is now fairly routinely performed in specialized labs within the field.

Major concerns:

Glycophenotypic bias of investigated neutrophils: The isolation of neutrophils was inappropriately performed using a surface glycoepitope CD15 (Lex) rather than the more common proteinaceous epitopes of CD11b, CD16 and CD66b etc or indeed density isolation. This is a particularly critical issue for this glycoproteomics manuscript since this approach risks introducing a significant glycophenotype bias in the population of neutrophils investigated. The purity of the isolated neutrophils was claimed to be 98% (although no details were provided of how this value was obtained), but it is unclear how many neutrophils were missed in the isolation process (and indeed if these carried a different glycophenotype). Neutrophils do not constitutively express Lex and sialyl Lex so a subpopulation has likely been studied. To this end, this glycoproteome profiling cannot be considered unbiased and need to be presented as the CD15+ neutrophil glycoproteome or similar. Some of the unusual signatures reported by this work may relate to this bias. It is expected that this serious shortcoming should be already made clear to readers in the abstract.

Inference of granule localization: The manuscript relies entirely on previous papers (i.e. Rorvig, JLB, 2013) to determine granule localization of the identified glycoproteins. This dependence on past literature in order to bring home one of the main conclusions of the study is not appropriate and the indirect line of evidence was not even mentioned in the abstract. A greater concern is that most, if not all, granule glycoproteins are actually distributed across all granules (not exclusive to any granule) as pointed out in Rorvig 2013. Even the most abundant azurophilic granule proteins are also found in the other granules and vice versa. The authors have ignored this important point and with their biased approach are unable to accurately establish the granule localization of the identified glycoproteins. The issue here is that their main conclusions build on this fundamental flaw. The title should be modified to reflect this uncertainty and the use of inference should be made clear to the readers throughout the manuscript including in the abstract.

Arbitrary classification of observed proteins: The identified glycoproteins were confusingly classified with respect to their location and/or subcellular location as inferred from databases. Examples include P5: "we could infer that our detections included proteins from the membrane, cytosol, azurophilic (primary) granules, specific (secondary) granules....", membrane proteins are present in all granule fractions including all other intracellular organelles, so it is unclear why this should form a separate category. Seems that all identified proteins previously reported from a granule origin could be better divided into a luminal/soluble and membrane sub-class. In Fig 5 it is confusing that the membrane proteins were divided into extracellular and intracellular membrane proteins. Should this instead be called plasma membrane proteins versus membranous proteins from intercellular organelles? Further in Fig 5: Why were the azurophilic proteins forming a separate category when the three other granules formed a single class? Why not separate the latter further? Moreover, "Secretory proteins" were mentioned in other parts of the manuscript and it is unclear how these differ from other identified luminal/soluble proteins from internal organelles. P13: "The proteins likely secreted from specific (secondary) granules, gelatinase (tertiary) granules and other secretory vesicles showed primarily diantennary and extended species". What is meant by "secreted"? The authors did not identify

proteins secreted via degranulation so it may be assumed that instead this wording is used to describe "soluble/luminal proteins". The authors should more logically re-classify the identified proteins and do more meaningful and easier-to-understand comparisons between those.

Inference of glycan structures: The study uses an analytical technique that does not determine the glycan fine structures but only the monosaccharide constituents of the glycans with site information. This aspect is mentioned in passing in the discussion but is not clear to the general readers throughout the manuscript and also not mentioned in the abstract. More concerning, confidently drawn glycans were depicted in Fig 3, Fig 4, and Fig 6 without the required structural support. Most glycans appear in many isomeric forms yet the indicated structures were all depicted with a clear topology without data support. The biosynthetic rules are not able to eliminate this uncertainty without some glycan fine structure data. All glycan structures (better written as generic compositions i.e. Hex, HexNAc, NeuAc, dHex) must therefore be indicated according to the structural evidence that was obtained. This is best done by either leaving out the cartoons altogether, drawing structures that indicate the structural ambiguity via brackets or just mentioning the generic monosaccharide compositions. The description of observed compositions on P10 seems more appropriate and should be used throughout. Figure 6 goes one step further; here the authors make a guess as to what glycan fine structures decorate each protein site in order to model these glycoproteins via available glycoprotein modelling software. Energy minimization was even applied, but again no evidence supporting the selected glycan fine structures could be provided making the efforts rather meaningless in my view.

Inaccurate glycan nomenclature: The manuscript uses ambiguous glycan nomenclature that does not follow conventions in this field making it difficult and confusing to read. Examples include "Hybrid/asymmetric" unclear what asymmetric means, not a phrase commonly used in glycobiology, "diantennary", this should be biantennary and does this cover more branched glycans as well? "extended", confusing naming since this indicates that these would only cover LacNAc extensions but are seemingly also covering highly branched tri- and tetra-antennary and bisecting glycans. High mannose should be oligomannose. The very truncated chitobiose core glycans were classified as paucimannose despite not having any terminal mannose which is also confusing. Some of the definitions of glycan classes on P7 were redundant. The authors should reclassify their observed glycan compositions in a more logical classification system that follows the conventions in field e.g. classify using common glycan types (oligomannose, hybrid, complex, paucimannose, chitobiose core...). The left hand side of the Figure 1 is educational and can be removed since this is better for text-books and the right hand side of that figure shows the simple work flow employed, which I do not find relevant since this single-approach study should be understandable without such helper graphics.

Glycopeptide bias and uncertainty: The glycopeptides were enriched using an unusual HILIC (cHILIC) phase not commonly used in glycopeptide profiling (ZIC-HILIC) bearing a distal cationic functional group instead of the usual anionic distal group. This manuscript reports on a high level of what are assumed to be M6P-glycans that are likely favoured in the enrichment step as shown with the comparison between the enrichment/no enrichment. Another potential concern in this regard is the EThcD triggering based on the 243 phosho-Man as one of a few other fragment ions. If the spectral counting included EThcD spectra (a detail that was not described (p22): "PSMs of individual glycan compositions were counted using an in-house script") this triggering strategy would favour the phosphomannose glycans. The integration and merging of spectral counting data between the different sample handling approaches (with and without enrichment) was not clearly described. Further, very few assigned glycopeptide spectra were provided (Fig 3) which is inappropriate when reporting on "unusual" structures. A concern is that many of the phosphomannose glycan compositions are very close in molecular mass to other non-phosphomannose glycans (0-2 Da) which make these impossible to discriminate with accurate MS1 measurements, hence quality MS2 spectra are required. Can the authors please ensure that the diagnostic 243 fragment ion is present at high mass accuracy in the phosphomannose spectra (HCD) they are reporting on?

Referencing: The manuscript throughout cites dated neutrophil and M6P literature (for example, Ref 1-7 are all 10-30 years old, Ref 47-48, 50 are 20-30 years old). Further, several key glycoanalytical papers were either not mentioned (for example PMID: 18587645 which was the first to profile neutrophil glycans) or not cited accurately (for example PMID: 32665399 was the first to precisely characterise glycans across the neutrophil granules). In all cases, this study did not appropriately compare their reported structural findings to the existing literature – the current manuscript only briefly mentioned congruence in the result section (p5) “The identified proteome was in congruence with those from other neutrophil proteomics studies...” but a much deeper comparison can be expected for both the proteins and glycan compositions observed when existing literature at the glycoprotein and glycome level is in fact available to enable such comparisons. For example, for the discussion of MPO (P9), a more exact and detailed comparison to the MPO literature could be expected, I think there are 3-4 papers out there on that topic. The neutrophil elastase profile on Asn124 showed an unexpected distribution not discussed. P1: “So far, a handful neutrophil proteins were reported to be decorated with atypical small glycans (paucimannose) and phosphomannosylated glycans”. The literature in fact describes more proteins with these modifications, which the authors should acknowledge. P4: “Here, by means of the latest advancements in sample preparation, hybrid mass spectrometry (MS) and data analysis²⁴⁻²⁶”, what method developments in these papers/reviews used in this manuscript are cited here and which sample prep and data analysis parts are referred to? Ion pairing HILIC? Reference 12 was not complete. Ref 51 outdated, new literature on this available. Inconsistencies in references (journal names missing or not short-hand format, some have all author names, some don’t).

O-glycosylation: Besides a single brief mention in the Figure 3 legend, no mention of O-glycans were made in the manuscript. The sample handling and data processing employed in this study would very likely enable O-glycopeptides to be present in the search engine output if these modifications exist on sequon-containing peptides, which is not that unlikely. Others have reported on neutrophil O-glycans, some of which can readily be mistaken as an N-glycan or as belonging to a co-existing N-glycan moiety if insufficiently resolved by EThcD including HexNAc1 (N-GlcNAc or O-GalNAc, both are expressed by neutrophils, in fact HexNAc1 is the most common glycan according to Figure 4), in particular, if such species co-exist on the same peptide. The common sialyl O-glycan can be mis-identified as an additional sialyl-lacNAc antenna of N-glycans etc. The authors are requested to appropriately address these O-glycosylation issues and inform the reader of how N- and O-glycans were confidently discriminated on the reported peptides (Asn resolution of the EThcD fragmentation or GlcNAc/GalNAc oxonium ion ratio) or where ambiguity exists.

Other issues:

1) Many ambiguous sentences, phrasing issues, inaccurate terminology, incorrect details including but not limited to: P1: “cause off-target damage”, do we really know what are on- and off-targets of neutrophils? These phrases are often used for drug treatment and gene editing etc but are not appropriate to use when describing our immune system. P2: “uncovering their unique glycosylation characteristics”, what does unique mean in this context. Both phosphomannose and paucimannose glycans have also been observed in other cell types including macrophages, stem cells and cancer cells which should be acknowledged in the manuscript. P2: “site-specific glycoproteomics”, glycoproteomics is by definition site-specific. P3: “The cells are unfortunately not very discriminatory..”, same phrasing issue as above. P3: “handling pathogens”, unclear, please rephrase. P3: “Even while”, please rephrase. P3: “directed modification”, please rephrase. P3: “can then be remodelled into a large degree of functionally-relevant variation”, unclear sentence. P4: “points towards the azurophilic granules contain at least paucimannosylation via a transcription-based timing mechanism, but phosphomannose glycans remain challenging to localize.....” unclear sentence and seemingly inaccurately cited, what does “transcription-based timing” mean and how did the paper show that (I assume 32665399 and 8870657 mean PMID 32665399 and PMID 8870657, why were these references not listed in the reference list?), what does “challenging to localize” mean”? P6: “with as notable examples” unclear. P7: “monoantennary and hybrid species (e.g. HexNAc3Hex3NeuAc1....)”

should this be Hex4?. P7: "phospho" should be phos, and unclear why "HexNAc4Hex8Phospho2" is mentioned, is this because the "uncapped" GlcNAc cappings of the M6P glycans were also considered? P8: "method to be leading in the detection of peptide glycoforms", unclear and similar issue elsewhere on P30 legend ("dominant"). P8: What does "visualization" mean in this context? P8: "we could predict their origins to be either from the cell surface", why was the azurophilic granule not mentioned in this sentence and how was the protein localization within the cell (ER/Golgi) determined from Ref 5 as cited. P9: "per glycoprotein each distinct...", unclear please rephrase. P11: "MS2-based quantitation" this means something else in quantitative proteomics, avoid wording and stick with spectral counting or PSM counting. P11: "HILIC is a commonly used for...", remove a, and not accurately phrased since cHILIC SPE not widely used for glycopeptide enrichment was used in this work. P13: "we highly recommend to isolate single proteins and match the findings from bottom-up MS with intact/native MS as described previously", but the authors at the same time report that this can be problematic PMID: 33856811. P14: "Lysosomal alpha-mannosidase (MAN2B1) can cause the removal of the alpha-linked mannoses that occupy most high-mannose and hybrid glycans", not correct, other alpha-mannosidases trim Man of oligomannose and hybrid type glycans. P15: "at Asn129 (monosaccharide)" what does this mean? P15: "from an other assembly", should this be "another" instead? P15: "Most cells can target glycoproteins for degradation through phosphorylation", unclear sentence, cells tag protein- and sugar-degrading hydrolases to lysosomes via M6P tagging. P16: "indivual", typo. P16: "Next to characterising the glycans on membrane-bound and secreted proteins", inaccurate. This study did not do glycan characterization and also did not study the glycosylation of secreted proteins. P17: "blood from 10 healthy donors", how much blood was actually collected and processed and was ethics required? P20: "Homo sapiens" in italics. P31: "are listed separately the other", unclear sentence.

2) Figures were not really discussed in text, just mentioned en passant. For example (but not limited to) the glycosylation in Fig 2 was not discussed at all. Several similar cases were found for other figures. Figure 3 legend did not mention what type of fragmentation method these spectra arise from.

3) Figure labels too small: Important labels and other details in Figure 4 and Figure 6 could not be read since they were too small. All labels need to be legible, otherwise these labels should be left out.

4) Seems odd to wait until discussion P10 to introduce the research questions. Should be moved to the end of the introduction to set the scene for the study.

5) P12: LacNAc extensions vs antennary branching are discussed but the existing literature that reports on this and other glycan structural motifs as determined by glycomics were not acknowledged (e.g. PMID: 18587645).

6) P14: "alternative explanation is that the glycosylation status of secretory proteins is highly dependent on the structure of the proteins themselves." Protein structural features impacting the extent of the glycan-processing have already been determined years ago and should be acknowledged. Similar issue with other "hypotheses" presented in the discussion. Literature exists for most of the hypotheses and proposed mechanisms and relationships mentioned in this manuscript and should be appropriately cited.

7) Experimental issues: Many missing details. The experiments should be described in greater details so they can be replicated by readers which is presently not the case. This issue relates to all parts of the experiment from sample prep over data collection to data analysis (ID and quant). P18: "3 μ m ZIC-cHILIC beads", what was the pore size? P19: "the glycopeptides were first eluted with 65% ACN/0.5% TFA, followed by 55% ACN/0.5% TFA." Seems like a small difference between the step wise elution conditions. Is there a risk that some glycopeptides were not eluted at the relatively high concentration of 55% ACN? P20: "approximately 300 ng (nonenriched) or 100 μ g (HILIC enriched) peptides were ionized..." Seems that an extreme amount of enriched glycopeptides was introduced into the mass spec. P20: "maximum injection time of 50 ms with a normalized collision energy of

27%." P20: "activation of ETD and supplemental activation with a normalized collision energy (NCE) of 27%." Supplementary energy seems high, please check if this is correct. P21: Need to clearly describe the search strategy for the glycoproteomics data analysis including the protease specificity and number of missed cleavages (some of these crucial details could only be inferred by the reader based on the supplementary table). P21: "Based on prior established curation criteria³³, we rejected glycopeptide identifications with a log probability lower than 1, and maintained a score threshold of 150", was no manual filtering performed? Figure 1 says that "curation" was performed but no manual filtering was mentioned in the method section. Expert filtering is still required to weed out false IDs from Byonic output. P22: Explanation is required about how membrane proteins were determined, unclear which database was used or if transmembrane prediction helix calculation was used. P22: "These criteria retained 85 reverse (glyco)peptide PSMs out of a total of 15095, establishing the FDR to be <1%." This is only relevant to the peptide part which should be clarified, the glycan can still be wrong.

8) Short title (P1): "Site-specific neutrophil N-glycoproteomics" – glycoproteomics is by definition site-specific.

Reviewer #2 (Remarks to the Author):

The authors have mostly answered to this Reviewer's comments and/or concerns. The inclusion of Fig S1 is well appreciated. However, for panel (D), the Figure legend says that m/z 657 is HexNAc1Hex1Fuc1NeuAc1, which is wrong. This may be a simple unintentional error but must be corrected. Also, the cartoon annotation seems to imply that m/z 657 resulted from loss of the Fuc from sLeX, or m/z 512 as derived from loss of the NeuAc from sLeX. This may indeed be the case but can equally be sialyl LacNAc on one antenna and LeX on the other. In fact, the only evidence indicating the presence of sLeX along with non-sialylated LacNAc, is the oxonium ion at m/z 803. Both isomeric structures could well co-exist and may or may not be resolved by the LC.

Reviewer #3 (Remarks to the Author):

The authors have answered all of my questions and the manuscript can be accepted for publication.

Response to Reviewers' comments:

Reviewer #1 (Remarks to the Author):

The authors have submitted a lightly revised R1 manuscript. After another detailed read, it is still my view that this manuscript, which applies a single line of enquiry and which relies on inference rather than direct evidence to reach biological conclusions (many of which have been shown before with a greater level of evidence) is better suited in a lower ranking specialist journal. Moreover, multiple critical issues were identified as detailed below. Finally, to put the study and its findings in a perspective, I think it deserves to be mentioned that a conventional glycoproteomics experiment as performed with this work is now fairly routinely performed in specialized labs within the field.

Answer: We acknowledge the time spent by the Reviewer as well as his/her expertise on the subject of glycoproteomics, however, we respectfully disagree with his/her view on the presented work. We also stand by our observation that neutrophil proteins dominantly found in azurophilic granules are distinctively glycosylated with pauci- and phosphomannose species. We believe this is an important finding that has been sufficiently substantiated in our manuscript, and have yet to see the routine application of the employed methodologies.

Major concerns:

Glycophenotypic bias of investigated neutrophils: The isolation of neutrophils was inappropriately performed using a surface glycoepitope CD15 (Lex) rather than the more common proteinaceous epitopes of CD11b, CD16 and CD66b etc or indeed density isolation. This is a particularly critical issue for this glycoproteomics manuscript since this approach risks introducing a significant glycophenotype bias in the population of neutrophils investigated. The purity of the isolated neutrophils was claimed to be 98% (although no details were provided of how this value was obtained), but it is unclear how many neutrophils were missed in the isolation process (and indeed if these carried a different glycophenotype). Neutrophils do not constitutively express Lex and sialyl Lex so a subpopulation has likely been studied. To this end, this glycoproteome profiling cannot be considered unbiased and need to be presented as the CD15+ neutrophil glycoproteome or similar. Some of the unusual signatures reported by this work may relate to this bias. It is expected that this serious shortcoming should be already made clear to readers in the abstract.

Answer: While CD15 is indeed the sugar Lewis X, and any selection procedure is expected to lead to some degree of bias, we see no reason to believe that our isolation on CD15 has captured anything other than the general population of neutrophils. Contrary to what the Reviewer states, CD15 is a constitutive component of neutrophils, for example demonstrated by Nakayama F et al, 2001, J Biol Chem (PMID: 11278338) and across several neutrophil subpopulations (including mature) by Evrard M et al, 2018, Immunity (PMID: 29466759).

We have added the above information to the revised manuscript. To additionally better inform on our FACS strategy and our assessment of the population purity, we have now included the FACS sorting as Figure S4 in the Supporting Information.

Inference of granule localization: The manuscript relies entirely on previous papers (i.e. Rorvig, JLB, 2013) to determine granule localization of the identified glycoproteins. This dependence on past literature in order to bring home one of the main conclusions of the study is not appropriate and

the indirect line of evidence was not even mentioned in the abstract. A greater concern is that most, if not all, granule glycoproteins are actually distributed across all granules (not exclusive to any granule) as pointed out in Rorvig 2013. Even the most abundant azurophilic granule proteins are also found in the other granules and vice versa. The authors have ignored this important point and with their biased approach are unable to accurately establish the granule localization of the identified glycoproteins. The issue here is that their main conclusions build on this fundamental flaw. The title should be modified to reflect this uncertainty and the use of inference should be made clear to the readers throughout the manuscript including in the abstract.

Answer: Rorvig S et al, 2013, J Leukoc Biol, indeed demonstrate that the granule location of proteins is not discrete, and that proteins can be present in multiple granules. However, in most cases there is a clear dominant granule for each protein, which we have selected for the comparison. Azurophilic granules, for instance, contain 70% of the myeloperoxidase, 84% of the azurocidin, 80% of the proteinase 3 (myeloblastin), 74% of the cathepsin G, and so forth (Rorvig S et al, 2013, J Leukoc Biol; PMID: 23650620). It is sensible to argue that the dominant glycan distribution of a protein would overlap with its dominant localization, particularly in case of abundances greater than 50%.

We have added the above information to the discussion of the manuscript so the reader can appropriately judge the process and have adjusted some of the working in the abstract to unambiguously reflect the content. However, we remain confident in our finding that proteins dominant in the azurophilic granules have markedly different glycosylation patterns than those from other subcellular locations.

Arbitrary classification of observed proteins: The identified glycoproteins were confusingly classified with respect to their location and/or subcellular location as inferred from databases. Examples include P5: “we could infer that our detections included proteins from the membrane, cytosol, azurophilic (primary) granules, specific (secondary) granules.....”, membrane proteins are present in all granule fractions including all other intracellular organelles, so it is unclear why this should form a separate category. Seems that all identified proteins previously reported from a granule origin could be better divided into a luminal/soluble and membrane sub-class. In Fig 5 it is confusing that the membrane proteins were divided into extracellular and intracellular membrane proteins. Should this instead be called plasma membrane proteins versus membranous proteins from intercellular organelles? Further in Fig 5: Why were the azurophilic proteins forming a separate category when the three other granules formed a single class? Why not separate the latter further? Moreover, “Secretory proteins” were mentioned in other parts of the manuscript and it is unclear how these differ from other identified luminal/soluble proteins from internal organelles. P13: “The proteins likely secreted from specific (secondary) granules, gelatinase (tertiary) granules and other secretory vesicles showed primarily diantennary and extended species”. What is meant by “secreted”? The authors did not identify proteins secreted via degranulation so it may be assumed that instead this wording is used to describe “soluble/luminal proteins”. The authors should more logically re-classify the identified proteins and do more meaningful and easier-to-understand comparisons between those.

Answer: Intra- and extracellular membrane proteins include those that are not explicitly present in one of the other vesicles, and have been categorized on basis of their general locations reported in the highly curated depository neXtprot. Proteins in “secretory vesicles” were assigned according to Rorvig et al, and differ in that respect from the proteins from the primary, secondary (specific) and tertiary (gelatinase) granules. In Figure 5, proteins from the secondary/tertiary granules and

secretory vesicles were grouped for visualization purposes, as we could detect no distinguishing or unique glycosylation features between them. Since pauci- and phosphomannose glycosylation could almost exclusively be detected for proteins dominantly present in the azurophilic granules, we opted to visualize this group separately.

Inference of glycan structures: The study uses an analytical technique that does not determine the glycan fine structures but only the monosaccharide constituents of the glycans with site information. This aspect is mentioned in passing in the discussion but is not clear to the general readers throughout the manuscript and also not mentioned in the abstract. More concerning, confidently drawn glycans were depicted in Fig 3, Fig 4, and Fig 6 without the required structural support. Most glycans appear in many isomeric forms yet the indicated structures were all depicted with a clear topology without data support. The biosynthetic rules are not able to eliminate this uncertainty without some glycan fine structure data. All glycan structures (better written as generic compositions i.e. Hex, HexNAc, NeuAc, dHex) must therefore be indicated according to the structural evidence that was obtained. This is best done by either leaving out the cartoons altogether, drawing structures that indicate the structural ambiguity via brackets or just mentioning the generic monosaccharide compositions. The description of observed compositions on P10 seems more appropriate and should be used throughout. Figure 6 goes one step further; here the authors make a guess as to what glycan fine structures decorate each protein site in order to model these glycoproteins via available glycoprotein modelling software. Energy minimization was even applied, but again no evidence supporting the selected glycan fine structures could be provided making the efforts rather meaningless in my view.

Answer: We believe we have been fair in highlighting the compositional nature of our observations throughout the manuscript, and we already discuss the structural certainties/uncertainties of our study at some length in the discussion (starting on page 12). In terms of glycan visualization, we find that brackets and other symbols of uncertainty make the cartoons harder to interpret than necessary, often obscuring the compositional information. Rather, we follow the example, set by others, to provide our best estimate of the glycosylation and to discuss via the main text and figure captions which limitations apply to the interpretation thereof. Publications that share our style of representation include Madunic K, et al, 2021, Anal. Chem., Blundell P, et al, 2019, J Immunol, and Zhang S, 2019, Mol Cell Proteomics - all recent publications from a diverse set of authors. The same reasoning applies to the structural models: we much prefer the visualization of the relative sizes of the glycan and protein moieties with its relative uncertainty, than no visualization at all.

Inaccurate glycan nomenclature: The manuscript uses ambiguous glycan nomenclature that does not follow conventions in this field making it difficult and confusing to read. Examples include "Hybrid/asymmetric" unclear what asymmetric means, not a phrase commonly used in glycobiology, "diantennary", this should be biantennary and does this cover more branched glycans as well? "extended", confusing naming since this indicates that these would only cover LacNAc extensions but are seemingly also covering highly branched tri- and tetra-antennary and bisecting glycans. High mannose should be oligomannose. The very truncated chitobiose core glycans were classified as paucimannose despite not having any terminal mannose which is also confusing. Some of the definitions of glycan classes on P7 were redundant. The authors should reclassify their observed glycan compositions in a more logical classification system that follows the conventions in field e.g. classify using common glycan types (oligomannose, hybrid, complex, paucimannose, chitobiose core...). The left hand side of the Figure 1 is educational and can be removed since this is better for text-books and the right hand side of that figure shows the simple work flow employed, which I do not find relevant since this single-approach study should be understandable without such helper graphics.

Answer: We do not consider our glycan nomenclature to be inaccurate or nonconventional, and follow examples already set by literature, *e.g.*, de Haan N, et al, 2020, Nat Rev Chem and Han J, 2020, J Prot. We are aware that different schools exist for glycan nomenclature and have therefore been deliberate in our choices. For example, “diantennary” is consistent with the Greek style of counting antennarities (di-, tri-, tetra-), whereas “biantennary” would make it the odd one out. “High mannose” and “oligomannose” are generally used interchangeably (even within the same article), and are also introduced alongside each other in important textbooks like Essentials of Glycobiology (3rd edition). We already mostly used “paucimannose” in the context of “paucimannose and smaller”, and have now made sure this is regularly mentioned throughout the manuscript.

Given the variety of glycan nomenclature throughout literature, we have made sure to precisely define what we mean by our classification, for example in the caption of Figure 4 and in the results section (page 7). Figure 1 will only aid in harmonizing the nomenclature and understanding of our manuscript, and we advocate its inclusion.

Glycopeptide bias and uncertainty: The glycopeptides were enriched using an unusual HILIC (cHILIC) phase not commonly used in glycopeptide profiling (ZIC-HILIC) bearing a distal cationic functional group instead of the usual anionic distal group. This manuscript reports on a high level of what are assumed to be M6P-glycans that are likely favoured in the enrichment step as shown with the comparison between the enrichment/no enrichment. Another potential concern in this regard is the EThcD triggering based on the 243 phospho-Man as one of a few other fragment ions. If the spectral counting included EThcD spectra (a detail that was not described (p22): “PSMs of individual glycan compositions were counted using an in-house script”) this triggering strategy would favour the phosphomannose glycans. The integration and merging of spectral counting data between the different sample handling approaches (with and without enrichment) was not clearly described. Further, very few assigned glycopeptide spectra were provided (Fig 3) which is inappropriate when reporting on “unusual” structures. A concern is that many of the phosphomannose glycan compositions are very close in molecular mass to other non-phosphomannose glycans (0-2 Da) which make these impossible to discriminate with accurate MS1 measurements, hence quality MS2 spectra are required. Can the authors please ensure that the diagnostic 243 fragment ion is present at high mass accuracy in the phosphomannose spectra (HCD) they are reporting on?

Answer: Our approach did use ZIC-HILIC beads (ZIC-cHILIC), as reported on page 18, which contains a zwitterionic functional group and should therefore be in line with commonly used HILIC stationary phases. As such, we expect no particular bias when comparing our study to others, aside from the bias we demonstrate, *e.g.*, in Figure 4. Secondly, a mass difference of 0-2 Da does not represent the accuracy and precision of our glycoproteomics findings, as these data were searched with a precursor mass tolerance of 10 ppm and a fragment mass tolerance of 20 ppm (page 21). For a typical 3000 Da precursor this would translate into a maximum error of ± 0.03 Da, and for the 243 fragment to a maximum error of approximately ± 0.005 Da. As the glycan masses closest to phosphomannosylated glycans are 2.12 Da apart (2x phospho compared to 1x hexose), wrongful assignment, even from a MS1 level, should be negligible. Lastly, while any triggering strategy might slightly skew results towards the triggers in question, omitting 243 would equally bias towards non-detection of the phosphomannose residues. As it is, given the mass precision of the detection as well as clear examples of phosphomannose fragmentation in Fig. 3 and Fig. S1 (HCD and EThcD fragmentation respectively), we see no reason to discount our phosphomannose detection as a technical artefact.

Referencing: The manuscript throughout cites dated neutrophil and M6P literature (for example, Ref 1-7 are all 10-30 years old, Ref 47-48, 50 are 20-30 years old). Further, several key glycoanalytical papers were either not mentioned (for example PMID: 18587645 which was the first to profile neutrophil glycans) or not cited accurately (for example PMID: 32665399 was the first to precisely characterise glycans across the neutrophil granules). In all cases, this study did not appropriately compare their reported structural findings to the existing literature – the current manuscript only briefly mentioned congruence in the result section (p5) “The identified proteome was in congruence with those from other neutrophil proteomics studies...” but a much deeper comparison can be expected for both the proteins and glycan compositions observed when existing literature at the glycoprotein and glycome level is in fact available to enable such comparisons. For example, for the discussion of MPO (P9), a more exact and detailed comparison to the MPO literature could be expected, I think there are 3-4 papers out there on that topic. The neutrophil elastase profile on Asn124 showed an unexpected distribution not discussed. P1: “So far, a handful neutrophil proteins were reported to be decorated with atypical small glycans (paucimannose) and phosphomannosylated glycans”. The literature in fact describes more proteins with these modifications, which the authors should acknowledge. P4: “Here, by means of the latest advancements in sample preparation, hybrid mass spectrometry (MS) and data analysis²⁴⁻²⁶”, what method developments in these papers/reviews used in this manuscript are cited here and which sample prep and data analysis parts are referred to? Ion pairing HILIC? Reference 12 was not complete. Ref 51 outdated, new literature on this available. Inconsistencies in references (journal names missing or not short-hand format, some have all author names, some don’t).

Answer: We are happy to include PMID: 18587645 in the introduction and have included an additional mention of PMID: 32665399. For the other references, while the majority thereof is from within the last decade, we generally do not want to discount relevant literature because of an age tag. As one example, Cieutat AM, et al., 1998, Blood, a reference that is 23 years old, still contains the most compelling histochemical evidence to date that azurophilic granules contain mannose-6-phosphate markers. It is work like this that had drawn our attention to the fact that mass spectrometric methods might have overlooked this important phenotype, which we now address in our manuscript. While we indeed compare several of our observations with the literature as verification on a technical level, we consider a protein-by-protein comparison to be outside of the scope of the manuscript. For this we envision/recommend a separate meta-analysis or review format. The reference format directly originated from EndNote using the “Nature” style and should be consistent in that regard, while the only book chapter (Ref 12) is now directly edited to show correctly.

O-glycosylation: Besides a single brief mention in the Figure 3 legend, no mention of O-glycans were made in the manuscript. The sample handling and data processing employed in this study would very likely enable O-glycopeptides to be present in the search engine output if these modifications exist on sequon-containing peptides, which is not that unlikely. Others have reported on neutrophil O-glycans, some of which can readily be mistaken as an N-glycan or as belonging to a co-existing N-glycan moiety if insufficiently resolved by EThcD including HexNAc1 (N-GlcNAc or O-GalNAc, both are expressed by neutrophils, in fact HexNAc1 is the most common glycan according to Figure 4), in particular, if such species co-exist on the same peptide. The common sialylT O-glycan can be mis-identified as an additional sialyl-lacNAc antenna of N-glycans etc. The authors are requested to appropriately address these O-glycosylation issues and inform the reader of how N- and O-glycans were confidently

discriminated on the reported peptides (Asn resolution of the EThcD fragmentation or GlcNAc/GalNAc oxonium ion ratio) or where ambiguity exists.

Answer: GlcNAc/GalNAc may indeed overlap with *O*-glycosylation, but we have provided several examples of clearly assigned Asn resolution in Fig. 3 and Fig S1 proving the existence of the heavily truncated *N*-glycans we discuss within the manuscript. Furthermore, most glycosylation sites display a distribution of glycans, including HexNAc1, HexNAc1dHex1, HexNAc2, HexNAc2dHex1, HexNAc2Hex1, HexNAc2Hex1dHex1, upwards to larger species. These composition groups would be difficult to acknowledge as *O*-glycosylation, but precisely fit the truncation process of an *N*-glycan, making the former very unlikely. Having said this, the combination of an *N*-glycan and an *O*-glycan on the same peptide, as the Reviewer mentions, would be extremely difficult to identify by a search engine or even by manual annotation, a process that yet needs to be introduced into the field. As such, the likelihood of a combination of *N*- and *O*-glycosylation on the same peptide remains to be investigated in general, but is not covered by the scope of this manuscript.

Other issues:

1) Many ambiguous sentences, phrasing issues, inaccurate terminology, incorrect details including but not limited to: P1: “cause off-target damage”, do we really know what are on- and off-targets of neutrophils? These phrases are often used for drug treatment and gene editing etc but are not appropriate to use when describing our immune system. P2: “uncovering their unique glycosylation characteristics”, what does unique mean in this context. Both phosphomannose and paucimannose glycans have also been observed in other cell types including macrophages, stem cells and cancer cells which should be acknowledged in the manuscript. P2: “site-specific glycoproteomics”, glycoproteomics is by definition site-specific. P3: “The cells are unfortunately not very discriminatory..”, same phrasing issue as above. P3: “handling pathogens”, unclear, please rephrase. P3: “Even while”, please rephrase. P3: “directed modification”, please rephrase. P3: “can then be remodelled into a large degree of functionally-relevant variation”, unclear sentence. P4: “points towards the azurophilic granules contain at least paucimannosylation via a transcription-based timing mechanism, but phosphomannose glycans remain challenging to localize.....” unclear sentence and seemingly inaccurately cited, what does “transcription-based timing” mean and how did the paper show that (I assume 32665399 and 8870657 mean PMID 32665399 and PMID 8870657, why were these references not listed in the reference list?), what does “challenging to localize” mean? P6: “with as notable examples” unclear. P7: “monoantennary and hybrid species (e.g. HexNAc3Hex3NeuAc1....)” should this be Hex4?. P7: “phospho” should be phos, and unclear why “HexNAc4Hex8Phospho2” is mentioned, is this because the “uncapped” GlcNAc cappings of the M6P glycans were also considered? P8: “method to be leading in the detection of peptide glycoforms”, unclear and similar issue elsewhere on P30 legend (“dominant”). P8: What does “visualization” mean in this context? P8: “we could predict their origins to be either from the cell surface”, why was the azurophilic granule not mentioned in this sentence and how was the protein localization within the cell (ER/Golgi) determined from Ref 5 as cited. P9: “per glycoprotein each distinct...”, unclear please rephrase. P11: “MS2-based quantitation” this means something else in quantitative proteomics, avoid wording and stick with spectral counting or PSM counting. P11: “HILIC is a commonly used for...”, remove a, and not accurately phrased since cHILIC SPE not widely used for glycopeptide enrichment was used in this work. P13: “we highly recommend to isolate single proteins and match the findings from bottom-up MS with intact/native MS as described previously”, but the authors at the same time report that this can be problematic PMID: 33856811. P14: “Lysosomal alpha-mannosidase (MAN2B1) can cause the removal of the alpha-linked mannoses that occupy most high-mannose and hybrid glycans”, not correct, other alpha-mannosidases trim Man of

oligomannose and hybrid type glycans. P15: “at Asn129 (monosaccharide)” what does this mean? P15: “from an other assembly”, should this be “another” instead? P15: “Most cells can target glycoproteins for degradation through phosphorylation”, unclear sentence, cells tag protein- and sugar-degrading hydrolases to lysosomes via M6P tagging. P16: “induvial”, typo. P16: “Next to characterising the glycans on membrane-bound and secreted proteins”, inaccurate. This study did not do glycan characterization and also did not study the glycosylation of secreted proteins. P17: “blood from 10 healthy donors”, how much blood was actually collected and processed and was ethics required? P20: “Homo sapiens” in italics. P31: “are listed separately the other”, unclear sentence.

Answer: We always appreciate suggestions to our writing and have adapted the manuscript where we considered appropriate. PMIDs 32665399 and 8870657 were indeed already included in the manuscript, but could be found in the document with accepted changes rather than the one with tracked changes; this is now consistent throughout the versions.

2) Figures were not really discussed in text, just mentioned en passant. For example (but not limited to) the glycosylation in Fig 2 was not discussed at all. Several similar cases were found for other figures. Figure 3 legend did not mention what type of fragmentation method these spectra arise from.

Answer: We believe to have discussed from each figure the evidence required to build the logical framework of the manuscript, but nevertheless the figures will always contain additional information that is of interest to the readers. To make the legends as clear as possible, we have now made sure that Fig. 3 is labelled as HCD fragmentation (and Fig. S1 as EThcD).

3) Figure labels too small: Important labels and other details in Figure 4 and Figure 6 could not be read since they were too small. All labels need to be legible, otherwise these labels should be left out.

Answer: We find legibility of labels an important consideration and did already make sure to test each figure by making a printout and receiving feedback from multiple people. While the labels are indeed on the small side, they nevertheless appear to be readable by our (non-expert) testers.

4) Seems odd to wait until discussion P10 to introduce the research questions. Should be moved to the end of the introduction to set the scene for the study.

Answer: We have now added a variant of our research questions to the introduction.

5) P12: LacNAc extensions vs antennary branching are discussed but the existing literature that reports on this and other glycan structural motifs as determined by glycomics were not acknowledged (e.g. PMID: 18587645).

Answer: We are happy to include reference PMID: 1858764 to acknowledge the glycan structural motifs and have done so in the discussion.

6) P14: “alternative explanation is that the glycosylation status of secretory proteins is highly dependent on the structure of the proteins themselves.” Protein structural features impacting the extent of the glycan-processing have already been determined years ago and should be acknowledged. Similar issue with other “hypotheses” presented in the discussion. Literature exists for most of the hypotheses and proposed mechanisms and relationships mentioned in this

manuscript and should be appropriately cited.

Answer: We have now included a highly cited reference covering the diverse functionalities of glycosylation, (namely, Varki A, 2016, Glycobiology), but generally consider our hypotheses to stem from interpretation of our results using common knowledge.

7) Experimental issues: Many missing details. The experiments should be described in greater details so they can be replicated by readers which is presently not the case. This issue relates to all parts of the experiment from sample prep over data collection to data analysis (ID and quant). P18: “3 μm ZIC-cHILIC beads”, what was the pore size? P19: “the glycopeptides were first eluted with 65% ACN/0.5% TFA, followed by 55% ACN/0.5% TFA.” Seems like a small difference between the step wise elution conditions. Is there a risk that some glycopeptides were not eluted at the relatively high concentration of 55% ACN? P20: “approximately 300 ng (nonenriched) or 100 μg (HILIC enriched) peptides were ionized...” Seems that an extreme amount of enriched glycopeptides was introduced into the mass spec. P20: “maximum injection time of 50 ms with a normalized collision energy of 27%.” P20: “activation of ETD and supplemental activation with a normalized collision energy (NCE) of 27%.” Supplementary energy seems high, please check if this is correct. P21: Need to clearly describe the search strategy for the glycoproteomics data analysis including the protease specificity and number of missed cleavages (some of these crucial details could only be inferred by the reader based on the supplementary table). P21: “Based on prior established curation criteria³³, we rejected glycopeptide identifications with a log probability lower than 1, and maintained a score threshold of 150”, was no manual filtering performed? Figure 1 says that “curation” was performed but no manual filtering was mentioned in the method section. Expert filtering is still required to weed out false IDs from Byonic output. P22: Explanation is required about how membrane proteins were determined, unclear which database was used or if transmembrane prediction helix calculation was used. P22: “These criteria retained 85 reverse (glyco)peptide PSMs out of a total of 15095, establishing the FDR to be <1%.” This is only relevant to the peptide part which should be clarified, the glycan can still be wrong.

Answer: We have gone through the suggested changes and amended the manuscript where appropriate. Glycoproteomics searches were performed with nonspecific digestion (explained on page 7, described in the methods on page 22), therefore protease specificity and the number of missed cleavages do not play a role. In general, we have made no textual errors in the methods section and the experiments were performed as described, in line with previous reports.

8) Short title (P1): “Site-specific neutrophil N-glycoproteomics” – glycoproteomics is by definition site-specific.

Answer: Fair enough, we have changed the short title to “Neutrophil N-glycoproteomics”.

Reviewer #2 (Remarks to the Author):

The authors have mostly answered to this Reviewer's comments and/or concerns. The inclusion of Fig S1 is well appreciated. However, for panel (D), the Figure legend says that m/z 657 is HexNAc1Hex1Fuc1NeuAc1, which is wrong. This may be a simple unintentional error but must be corrected. Also, the cartoon annotation seems to imply that m/z 657 resulted from loss of the Fuc

from sLeX, or m/z 512 as derived from loss of the NeuAc from sLeX. This may indeed be the case but can equally be sialyl LacNAc on one antenna and LeX on the other. In fact, the only evidence indicating the presence of sLeX along with non-sialylated LacNAc, is the oxonium ion at m/z 803. Both isomeric structures could well co-exist and may or may not be resolved by the LC.

Answer: Indeed, HexNAc₁Hex₁Fuc₁NeuAc₁ can be found at m/z 803.29, whereas m/z 657.23 corresponds to HexNAc₁Hex₁NeuAc₁ - we thank the reviewer for noticing. Isomeric structures are likely to coexist in our fragmentation spectra, and we have now made mention of this in the legend of Figure S1: "Note that the fragments above may have originated from a mixture of isomeric structures, wherein alternative fragmentation pathways could lead to observed ions like HexNAc₁Hex₁NeuAc₁ (m/z 657.23) and HexNAc₁Hex₁dHex₁ (m/z 512.20)."

Reviewer #3 (Remarks to the Author):

The authors have answered all of my questions and the manuscript can be accepted for publication.